

# Exploring DMS oxidation and implications for global aerosol radiative forcing

Ka Ming Fung[1], Colette L. Heald[1,2], Jesse H. Kroll[1], Siyuan Wang[3], Duseong S. Jo[4], Andrew Gettelman[4], Zheng Lu[5], Xiaohong Liu[5], Rahul A. Zaveri[6], Eric Apel[4], Donald R. Blake[7], Jose-Luis

Jimenez[8], Pedro Campuzano-Jost[8], Patrick Veres[3], Timothy S. Bates[9], John E. Shilling[10], Maria Zawadowicz[11]

[1]  Department of Civil and Environmental Engineering, Massachusetts Institute of Technology, Cambridge, MA, USA

[2]  Department of Earth, Atmospheric and Planetary Sciences, Massachusetts Institute of Technology, Cambridge, MA, USA

[3]  Chemical Sciences Laboratory, National Oceanic and Atmospheric Administration, Boulder, CO, USA

[4]  Atmospheric Chemistry Observations and Modeling Laboratory, National Center for Atmospheric Research, Boulder, CO, USA

[5]  Department of Atmospheric Sciences, Texas A&M University, College Station, TX, USA

[6]  Pacific Northwest National Laboratory, Richland, WA, USA

[7]  Department of Chemistry, University of California, Irvine, CA, USA

[8]  Department of Chemistry & Cooperative Institute for Research in Environmental Sciences, University of Colorado, Boulder, CO, USA

[9]  The Cooperative Institute for Climate, Ocean, and Ecosystem Studies, College of the Environment, University of Washington, Seattle, WA, USA

[10]  Atmospheric Sciences and Global Change Division, Pacific Northwest National Laboratory, Richland, WA, USA

[11]  Environmental and Climate Sciences Department, Brookhaven National Laboratory, Upton, NY, USA

*Corresponding to:* Ka Ming Fung (kamingfung@mit.edu) & Colette L. Heald (heald@mit.edu)



**Abstract.** Aerosol indirect radiative forcing (IRF), which characterizes how aerosols alter cloud formation and properties, is very sensitive to the preindustrial (PI) aerosol burden. Dimethyl sulfide (DMS), emitted from the ocean, is a dominant natural precursor of non-sea-salt sulfate in the PI and pristine present-day (PD) atmospheres. Here we revisit the atmospheric oxidation chemistry of DMS, particularly under pristine conditions, and its impact on aerosol IRF. Based on previous laboratory

studies, we expand the simplified DMS oxidation scheme used in the Community Atmospheric Model version 6 with chemistry (CAM6-chem) to capture the OH-addition pathway as well as the H-abstraction pathway and the associated isomerization branch. These additional oxidation channels of DMS produce several stable intermediate compounds, e.g., methanesulfonic acid (MSA) and hydroperoxymethyl thioformate (HPMTF), delay the formation of sulfate, and hence, alter the spatial

distribution of sulfate aerosol and radiative impacts. The expanded scheme improves the agreement between modeled and observed concentrations of DMS, MSA, HPMTF, and sulfate over most marine regions based on the NASA Atmospheric Tomography (ATom), the Aerosol and Cloud Experiments in the Eastern North Atlantic (ACE-ENA), and the VAMOS Ocean-Cloud-Atmosphere-Land Study Regional Experiment (VOCALS-REx) measurements. We find that the global HPMTF burden, as well

as the burden of sulfate produced from DMS oxidation are relatively insensitive to the assumed isomerization rate, but the burden of HPMTF is very sensitive to a potential additional cloud loss. We find that global sulfate burden under PI and PD emissions increase to 412 Gg-S (+29%) and 582 Gg-S (+8.8%), respectively, compared to the standard simplified DMS oxidation scheme. The resulting annual mean global PD direct radiative effect of DMS-derived sulfate alone is $-0.11$ W m$^{-2}$. The

enhanced PI sulfate produced via the gas-phase chemistry updates alone dampens the aerosol IRF as anticipated ($-2.2$ W m$^{-2}$ in standard versus $-1.7$ W m$^{-2}$ with updated gas-phase chemistry). However, high clouds in the tropics and low clouds in the Southern Ocean appear particularly sensitive to the additional aqueous-phase pathways, counteracting this change ($-2.3$ W m$^{-2}$). This study confirms the sensitivity of aerosol IRF to the PI aerosol loading, as well as the need to better understand the processes

controlling aerosol formation in the PI atmosphere and the cloud response to these changes.



## 1 Introduction

The IPCC AR5 (Myhre et al., 2013) indicates that atmospheric aerosol particles are a dominant source of uncertainty in global climate forcing. Aerosols interact with incoming and outgoing radiation
directly (via scattering and absorption) and indirectly (via changing cloud properties and lifetime). In particular, the aerosol indirect radiative forcing (IRF) via interactions with clouds is driven by the fractional enhancement of aerosol burden from a preindustrial (PI; 1850) atmosphere to a present-day (PD; 2000) one, with a cleaner PI atmosphere producing a larger IRF (Menon et al., 2002). Carslaw et al. (2013) confirm that the estimated uncertainty in aerosol IRF is dominated by uncertainty in natural
aerosols. It is therefore critically important to accurately determine the formation of natural aerosols and their radiative impacts in both PD and PI atmosphere.

Marine dimethyl sulfide (DMS: $CH_3SCH_3$) accounts for >50% of natural gas-phase sulfur emissions (Chin et al., 1996; Andreae, 1990; Kilgour et al., 2021). Once emitted into the troposphere, oxidation of DMS takes place within 1–2 days, forming other sulfur-containing products such as
sulfuric acid ($H_2SO_4$) and methane sulfonic acid (MSA: $CH_3SO_3H$) (Boucher et al., 2003; Breider et al., 2010). These gaseous products can facilitate the formation of new particles as well as cloud condensation nuclei (CCN), especially in the marine boundary layer (MBL) (Charlson et al., 1987; von Glasow and Crutzen, 2004; Kulmala et al., 2000). Sulfate and MSA formed in particle-phase can directly impact the size distribution of aerosols and alter cloud microphysics (Kaufman and Tanré,
1994). DMS is estimated to be responsible for up to 11–18% of global sulfate burden in PD (Yang et al., 2017; Gondwe et al., 2003) and >48% of atmospheric sulfur burden in PI (Tilmes et al., 2019). Though the crucial role of DMS oxidation as a source of natural aerosols has been acknowledged for decades, its oxidation mechanisms are still not well understood (Barnes et al., 2006; Hoffmann et al., 2016).

Global/regional models often simplify the DMS oxidation processes for the sake of computational costs. For example, the Community Atmosphere Model with chemistry (CAM-chem) includes only the oxidation of DMS by OH and $NO_3$ radicals, directly producing $SO_2$, which further



oxidizes to produce sulfate (Emmons et al., 2020; Lamarque et al., 2012). This simplification ignores some potentially important reaction intermediates and pathways. For instance, previous studies suggest

that BrO contributes up to 30% of the DMS sink in remote MBL (Boucher et al., 2003; Breider et al., 2010; von Glasow and Crutzen, 2004; Khan et al., 2016). MSA has been found to form efficiently via the multi-phase OH-addition DMS oxidation pathway followed by reaction with $OH_{(aq)}$ to form sulfate aerosol in the MBL (von Glasow and Crutzen, 2004; Milne et al., 1989; Zhu et al., 2006). Recently, both theoretical and laboratory studies have proposed that a pristine environment favors the H-

abstraction reaction when DMS is oxidized by OH, generating methylthiomethylperoxy radicals (MSP: $CH_3SCH_2OO$), which further undergoes a series of rapid intramolecular H-shift isomerization reactions, yielding a stable intermediate hydroperoxymethyl thioformate (HPMTF: $HOOCH_2SCHO$) (Wu et al., 2015; Berndt et al., 2019). Recent in situ measurements report HPMTF concentrations that equal or even exceed DMS concentrations in the MBL during the day, confirming the importance of the

isomerization branch for capturing the fate of oxidized DMS (Veres et al., 2020; Vermeuel et al., 2020).

The lifetimes of stable intermediates from DMS oxidation can be up to days. As a result, these intermediates can delay the formation of DMS-derived sulfate, affecting not only the spatial distribution of sulfate aerosols but also the effective sulfate yield from DMS as unreacted sulfate precursors may be subject to physical removal through wet or dry deposition. Thus, neglecting these intermediates could

lead to misrepresentation of the spatial distribution of sulfate aerosol loading and limit our ability to accurately determine aerosol radiative forcing.

Here, we implement a more detailed multi-generational and multi-phase chemical mechanism to describe DMS oxidation within the Community Atmosphere Model version 6 with chemistry (CAM6-chem) (Emmons et al., 2020) – the atmosphere component of the Community Earth System

Model version 2.1 (CESM2.1) (Danabasoglu et al., 2020). The expanded chemistry captures the formation of stable intermediates such as MSA and HPMTF alongside $SO_2$. We perform multiple sensitivity tests to investigate how the uncertainty in modeling the newly confirmed HPMTF could influence the DMS chemistry and the resulting atmospheric sulfate burden. The model results are compared against an array of in situ observations. Finally, we examine how the natural aerosol



background from DMS oxidation simulated with the modified model impacts estimates of aerosol radiative forcing.

## 2 Model description

CESM2.1 consists of model components that quantitatively describe the atmosphere, land, sea-
ice, land-ice, rivers, and ocean (Danabasoglu et al., 2020). Fluxes and state variables are exchanged through a coupler to describe the co-evolution of these earth system components. Here, we run with a coupled atmosphere (CAM6-chem) and land (Community Land Model, CLM5) and use prescribed data for the remaining earth system components. In particular, sea surface temperature (SST) and sea ice conditions (Hurrell et al., 2008), as well as the mixing ratios of greenhouse gases (Meinshausen et al.,
2017) are all fixed to present-day conditions. Following similar practices in previous studies (Gettelman, 2015; Gettelman et al., 2019), this configuration aims to constrain the potential environmental feedbacks such that the aerosol effects (on atmospheric composition, cloud, and radiation) are due to the change in emissions and chemistry only.

### 2.1 Model configuration

In this work, CAM6-chem is run in an online configuration with free dynamics at $1.9º \times 2.5º$ (latitude by longitude) horizontal resolution and 32 vertical layers (surface to 3 hPa or ~45 km), with a model timestep of 30 minutes. The default chemistry scheme is Model of Ozone and Related chemical Tracers with representations of both tropospheric and stratospheric chemistry (MOZART-TS1) (Emmons et al., 2020) with a Volatility Basis Set (VBS) scheme specifically for the gas-phase
intermediate semi-volatile organic precursors of secondary organic aerosols (SOA) (Tilmes et al., 2019). The DMS chemistry is described in further detail in **Sect. 2.3**. Aerosols are simulated using the Modal Aerosol Model with four modes (MAM4) (Liu et al., 2016) coupled with Model for Simulating Aerosol Interactions and Chemistry (MOSAIC) (Zaveri et al., 2008, 2021; Lu et al., 2021), for sulfate ($SO_4^{2-}$), ammonium ($NH_4^+$), nitrate ($NO_3^-$), primary organic matters (POM), SOA, sea salt, and mineral dust.



MAM4 classifies aerosols into three size-dependent modes (Aitken, accumulation, and coarse) with an additional primary carbon mode for handling the aging of fine POM and black carbon (BC). Size distributions of aerosols in each mode are assumed to be log-normal with fixed geometric standard deviations and varying mode dry or wet radius depending on particle number and changes in total dry or wet volume (Liu et al., 2012). MAM4 defines the cut-off size ranges of 0.015–0.053 μm for aerosol

in Aitken mode, 0.058–0.27 μm for accumulation mode, and 0.80–3.65 μm for coarse mode. Dynamic partitioning of $H_2SO_4$, $HNO_3$, HCl, and $NH_3$ to each mode, related particle-phase thermodynamics, as well as water content and pH of interstitial aerosols, are computed using the MOSAIC module (Zaveri et al., 2008, 2021; Lu et al., 2021). Further details describing the dry and wet deposition, aerosol optical properties, radiative transfer, and aerosol-cloud microphysics are described in **Supplementary**

**Information**; these processes are all based on the standard CAM6-chem.

We perform four sets of simulations for the PD and PI atmospheric conditions with the standard and our modified chemical schemes. Details of the runs are tabulated in **Table 1**. Each run is performed for 10 years with the first year as spin-up and averages over the latter nine years are presented in our results.


**Table 1.** Configuration of key simulation cases in this study

| Case Alias | Chemistry | Anthropogenic Emissions |
|---|---|---|
| STD_2000 | TS1 with the model-default DMS oxidation reactions in | 2000-level |
| STD_1850 | **Table 3** | 1850-level |
| | | |
| MOD_2000 | TS1 with the new gas-phase and aqueous-phase reactions | 2000-level |
| MOD_1850 | in **Table 4**, **Table 5**, **Table 6**, and **Table 7**. | 1850-level |
| | | |
| MOD_RE_2000 | | 2000-level |





| MOD_RE_1850 | TS1 with the new gas-phase and aqueous-phase reactions in **Table 4**, **Table 5**, **Table 6**, and **Table 7,** together with artificial rapid conversion of MSA to sulfate for assessing the radiative effect of MSA. | 1850-level |
| GAS_RE_2000 | TS1 with the new gas-phase reactions in **Table 4**, **Table** | 2000-level |
| GAS_RE_1850 | 5, and **Table 6**, together with artificial rapid conversion of MSA to sulfate for assessing the radiative effect of MSA | 1850-level |

## 2.2 Emissions

DMS is emitted from the ocean ($E_{DMS}$) via the Online Air-Sea Interface for Soluble Species (OASISS) model developed for CAM6-chem, which has been validated against observations for

acetaldehyde (Wang et al., 2019b), acetone (Wang et al., 2020), and organohalogens (e.g. $CHBr_3$ and $CH_2Br_2$) (Wang et al., 2019c). OASISS employs a two-layer framework that considers transfer velocities both through air and through water ($k_{air}$ and $k_{water}$) (Johnson, 2010):

$$E_{DMS} = k \left( [DMS]_{water} - \frac{[DMS]_{air}}{H_{DMS}} \right) (1 - f_{cice}) \tag{1}$$

$$k = \left( \frac{1}{k_{water}} + \frac{1}{k_{air}} \right)^{-1} \tag{2}$$

where $k$ (m s$^{-1}$) is the overall transfer velocity. The surface seawater concentration, $[DMS]_{water}$ (nM) is prescribed to follow the Lana et al. (2011) sea-surface DMS climatology in both our PI and PD simulations. DMS mixing ratio in the air, $[DMS]_{air}$, and its Henry's Law constant, are from CAM6-chem. $k_{air}$ is based on the NOAA COARE algorithm (Jeffery et al., 2010), which is a function of surface wind speed, with an additional adjustment for the diffusivity of still air (Mackay and Yeun, 1983). $k_{water}$

is based on Nightingale et al. (2000), which considers sea surface temperature and salinity. Lastly, $f_{cice}$





is the fraction of sea-ice coverage in each grid cell, such that DMS emission is suppressed from sea-ice covered surfaces.

On average, the global annual-total marine $E_{DMS}$ is 21.5 Tg-S yr$^{-1}$ in [STD_2000]. Meteorological variability has little impact on the interannual variability of emissions (<±4%). Our $E_{DMS}$ is higher than the 18 Tg-S yr$^{-1}$ than the model default inventory (Kettle and Andreae, 2000), but lower than the 28 Tg-S yr$^{-1}$ reported in the original model study by Lana et al. (2011) and within the range of 11–28 Tg-S yr$^{-1}$ estimates simulated by GEOS-Chem, TOMCAT-GLOMAP and other models (Lennartz et al., 2015; Spracklen et al., 2005; Hezel et al., 2011). The estimation of $E_{DMS}$ is sensitive to the choice of sea-surface DMS climatology; Chen et al. (2018) show that emissions vary from 18 Tg-S yr$^{-1}$ with the Kettle et al. (1999) DMS climatology to 22 Tg-S yr$^{-1}$ with the Lana et al. (2011) DMS climatology.

In all simulations, anthropogenic emissions are from the Community Emissions Data System (CEDS) (Hoesly et al., 2018) and biomass burning emissions from the CMIP6 inventory (van Marle et al., 2017). Biogenic emissions are estimated online from CLM5 using the Model of Emissions of Gases and Aerosols from Nature (MEGAN) version 2.1 (Guenther et al., 2012). CAM6-chem assumes that 2.5% by molar of sulfur emitted from the Energy & Industry sector is already in the form of primary sulfate aerosols (in accumulation mode). Volcanic emission inventories for $SO_2$ and primary sulfate are categorized by volcano types, i.e., continuously outgassing volcanos (Andres and Kasgnoc, 1998; Carn et al., 2017) and eruptive volcanoes (Mills et al., 2016). All volcanic emissions are fixed at the 2000-level in both PI and PD simulations. A breakdown of $SO_2$ emissions in this study is summarized in **Table 2**. We use the same emissions for other species as the standard CMIP6 simulations (Emmons et al., 2020).

**Table 2.** $SO_2$ emissions in this study.

| Sources | Annual Total (Tg-S) | References |
|---|---|---|



| | PI | PD | |
|---|---|---|---|
| **Total anthropogenic emission** | **1.14** | **54.2** | |
| Agriculture, solvents, and human waste | <0.01 | 0.2 | (Hoesly et al., 2018) |
| Residential and transportation | 0.4 | 5.2 | (Hoesly et al., 2018) |
| Shipping | 0.04 | 4.3 | (Hoesly et al., 2018) |
| Energy and industry | 0.7 | 44.4 | (Hoesly et al., 2018) |
| Aircraft | - | 0.1 | (Hoesly et al., 2018) |
| | | | |
| **Total natural emission** | **22.9** | **22.9** | |
| Biomass burning | 1.0 | 1.0 | (van Marle et al., 2017) |
| Volcanos | 21.9 | 21.9 | (Andres and Kasgnoc, 1998; Carn et al., 2017; Mills et al., 2016) |

## 185 2.3 Expanded DMS Oxidation Scheme

The standard CAM6-chem contains three gas-phase DMS oxidation reactions (**Table 3**) (Barth et al., 2000; Emmons et al., 2010). These reactions simplify the DMS oxidation chemistry by treating only gas-phase reactions and producing $SO_2$ directly, neglecting the role of multiphase chemistry and other key chemical products and intermediates found in chamber and field studies (e.g., Hoffmann et al., 2016; Wu et al., 2015). We note that the second reaction does not conserve sulfur.

**Table 3.** The three DMS oxidation reactions in the standard CAM6-chem.

| Gas-phase Reactions | $k_{298}$ (cm$^3$ molecule$^{-1}$ s$^{-1}$) | $-E_a/R$ (K) | References |
|---|---|---|---|
| DMS + OH → $SO_2$ (H-abstraction) | $9.60 \times 10^{-12}$ | –234 | (Emmons et al., 2010) |
| DMS + OH → $0.5SO_2 + 0.5HO_2$ (OH-addition) | See note[a] | | (Emmons et al., 2010) |
| DMS + $NO_3$ → $SO_2$ + $HNO_3$ | $1.90 \times 10^{-13}$ | 520 | (Emmons et al., 2010) |

[a] $k(T, [O_2], [M]) = 1.7 \times 10^{-42} e^{(7810/T)} \times 0.21[M] / (1 + 5.5 \times 10^{-31} e^{(7460/T)} \times 0.21[M])$ (cm$^3$ molecule$^{-1}$ s$^{-1}$)





To improve the representation of DMS oxidation in CAM6-chem, we add a suite of new reactions that describe the chemical evolution from DMS to $SO_2$, and ultimately, sulfate via the H-abstraction and OH-addition pathways. **Figure 1** illustrates the expanded chemistry schematically. Our additions are based on recent laboratory studies and field observations and are discussed in detail in

what follows.

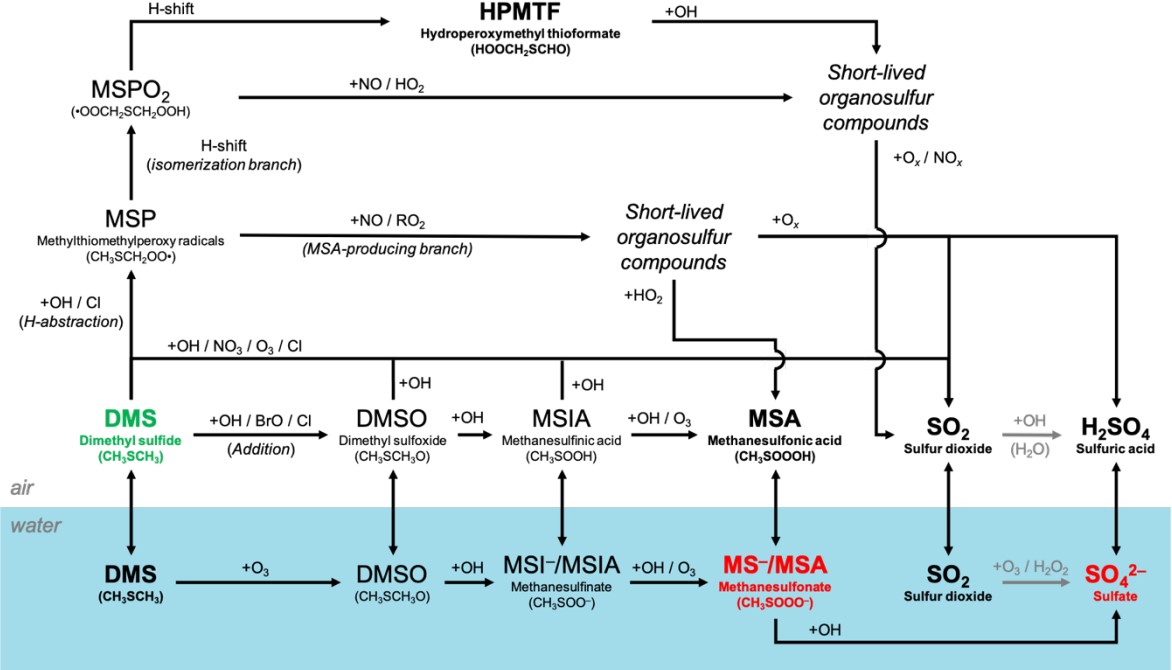

**Figure 1.** A schematic summary of our expanded chemistry of DMS oxidation in CAM6-chem (**Table 4**, **Table 5**, **Table 6**, and **Table 7**). Key relatively long-lived species (DMS, MSA, HPMTF, SO₂, and sulfate) are highlighted in bold. The blue shadings denote species and reactions in the aqueous phase in interstitial aerosols and cloud droplets. DMS (highlighted in green) can undergo O-atom addition (rightward path) or H-abstraction (upper paths). The H-abstraction channel further diverts into the isomerization branch (top path) and the MSA-producing branch. SO₂ is the dominant product of most gas-phase pathways while MSA is formed mainly via the aqueous-phase oxidation of DMS. Oxidation of SO₂ to sulfate or sulfuric acid is handled by the CAM6-chem standard chemistry. The resultant particulate MSA and sulfate (highlighted in red) are key species with important radiative impacts.





### 2.3.1 The H-abstraction Pathway

The H-abstraction reactions of DMS with OH or Cl generate MSP, which then either reacts with
NO or $RO_2$ forming of MSA and $H_2SO_4$, or undergoes consecutive intramolecular H-shift reactions
(isomerization), yielding HPMTF and $SO_2$. Hence, we group these two serial reactions into two
branches, namely the MSA-producing branch and the isomerization branch. The reactions of the MSA-
producing branch are tabulated in **Table 4**. These reactions are largely based on Hoffmann et al. (2016),
who combined the chemical mechanism from the Master Chemical Mechanism version 3 (MCM v3)
(Saunders et al., 2003) and other laboratory and computational studies. The reactions in the
isomerization branch are detailed in **Table 5**. Here, we use an observationally constrained isomerization
rate of MSP ($k_{iso}$) of 0.04 $s^{-1}$ at 293K estimated by Veres et al. (2020). This new $k_{iso}$ is slower than the
previously determined values of 0.23 $s^{-1}$ at 295K (Berndt et al., 2019) to 2.1 $s^{-1}$ at 293K (Wu et al.,
2015), delaying the formation of HPMTF. A recent chamber experiment estimates an intermediate $k_{iso}$
value of 0.12 $s^{-1}$ at 293K (Ye et al., 2021). We investigate the impact of the uncertainty in $k_{iso}$, in **Sect.
3**. The only chemical loss process of HPMTF in our model is oxidation by OH at a rate of $1.11 \times 10^{-11}$
$cm^3$ $molecule^{-1}$ $s^{-1}$ as recommended by Vermeuel et al. (2020), which is an experimentally determined
OH-oxidation rate of MTF (a structurally similar proxy to HPMTF) by Patroescu et al. (1996).
Oxidizing HPMTF at this rate was shown to match better with recent measurements (Vermeuel et al.,
2020) than the $1.4 \times 10^{-12}$ $cm^3$ $molecule^{-1}$ $s^{-1}$ suggested by a computational study (Wu et al., 2015).
Recent studies (Veres et al., 2020; Vermeuel et al., 2020) suggest that cloud uptake is another important
sink of HPMTF; we include a series of sensitivity tests based on [MOD_2000] to address the uncertainty
in the HPMTF budget arising from this potential loss process. Vermeuel et al. (2020) report that using
a cloud uptake rate ($k_{HPMTF+cloud}$) at $5 \times 10^{-3}$ $s^{-1}$ results in a better match of diurnal variability of HPMTF
with their local measurements. Due to the lack of detailed measurement, we use this $k_{HPMTF+cloud}$ and a
slower hypothetical value at $5 \times 10^{-5}$ $s^{-1}$ for our sensitivity tests.





**Table 4.** Summary of the MSA-producing branch of the H-abstraction pathway in the DMS chemistry implemented into CAM6-chem.

| Gas-phase Reactions | $k_{298}$ (cm$^3$ molecule$^{-1}$ s$^{-1}$) | $-E_a/R$ (K) | References |
|---|---|---|---|
| DMS + OH → MSP (CH$_3$SCH$_2$OO) | $1.12 \times 10^{-11}$ | –250 | (Saunders et al., 2003) |
| DMS + Cl → 0.45MSP + 0.55(CH$_3$)$_2$S(Cl) + 0.45HCl | $3.40 \times 10^{-10}$ | | IUPAC |
| (CH$_3$)$_2$S(Cl) → DMS + Cl | $9.00 \times 10^{1}$ | | (Enami et al., 2004) |
| MSP + NO → CH$_3$SCH$_2$(O) + NO$_2$ | $4.90 \times 10^{-12}$ | 260 | (Saunders et al., 2003) |
| MSP + RO$_2$ → CH$_3$SCH$_2$(O) + O$_2$ | $3.74 \times 10^{-12}$ | | (Saunders et al., 2003) |
| CH$_3$SCH$_2$(O) → CH$_3$S + CH$_2$O | $1.00 \times 10^{6}$ | | (Saunders et al., 2003) |
| CH$_3$S + O$_3$ → CH$_3$S(O) + O$_2$ | $1.15 \times 10^{-12}$ | 430 | (Saunders et al., 2003) |
| CH$_3$S + O$_2$ → CH$_3$S(OO) | $1.20 \times 10^{-16}$ | 1580 | (Saunders et al., 2003) |
| CH$_3$S(O) + O$_3$ → CH$_3$(O$_2$) + SO$_2$ | $4.00 \times 10^{-13}$ | | (Saunders et al., 2003) |
| CH$_3$S(OO) → CH$_3$(O$_2$) + SO$_2$ | $5.60 \times 10^{16}$ | –10870 | (Saunders et al., 2003) |
| CH$_3$S(OO) → CH$_3$SO$_2$ | $1.00$ | | (Saunders et al., 2003) |
| CH$_3$SO$_2$ + O$_3$ → CH$_3$SO$_3$ + O$_2$ | $3.00 \times 10^{-13}$ | | (Saunders et al., 2003) |
| CH$_3$SO$_2$ → CH$_3$(O$_2$) + SO$_2$ | $5.00 \times 10^{13}$ | –9673 | (Saunders et al., 2003) |
| CH$_3$SO$_3$ + HO$_2$ → MSA + O$_2$ | $5.00 \times 10^{-11}$ | | (Saunders et al., 2003) |
| CH$_3$SO$_3$ → CH$_3$(O$_2$) + H$_2$SO$_4$ | $5.00 \times 10^{13}$ | –9946 | (Saunders et al., 2003) |

**Table 5.** Summary of the isomerization branch of the H-abstraction pathway in the DMS chemistry implemented into CAM6-chem.

| Gas-phase Reactions | $k_{298}$ (cm$^3$ molecule$^{-1}$ s$^{-1}$) | $-E_a/R$ (K) | References |
|---|---|---|---|
| MSP → OOCH$_2$SCH$_2$OOH | See note[a] | | (Veres et al., 2020) |
| OOCH$_2$SCH$_2$OOH → HPMTF (HOOCH$_2$SCHO) + OH | See note[b] | | (Veres et al., 2020) |
| OOCH$_2$SCH$_2$OOH + NO → HOOCH$_2$SCH$_2$O + NO$_2$ | $4.90 \times 10^{-12}$ | 260 | (Saunders et al., 2003) |





| | | | |
|---|---|---|---|
| $HOOCH_2SCH_2O \rightarrow HOOCH_2S + CH_2O$ | $1.00 \times 10^6$ | | (Saunders et al., 2003) |
| $OOCH_2SCH_2OOH + HO_2 \rightarrow HOOCH_2SCH_2OOH + O_2$ | $1.13 \times 10^{-13}$ | 1300 | (Saunders et al., 2003) |
| $HPMTF + OH \rightarrow HOOCH_2SCO + H_2O$ | $1.11 \times 10^{-11}$ | | (Patroescu et al., 1996; Vermeuel et al., 2020) |
| $HOOCH_2SCO \rightarrow HOOCH_2S + CO$ | $9.20 \times 10^9$ | –505.4 | (Wu et al., 2015) |
| $HOOCH_2SCO \rightarrow OH + CH_2O + OCS$ | $1.60 \times 10^7$ | –1468.6 | (Wu et al., 2015) |
| $HOOCH_2S + O_3 \rightarrow HOOCH_2SO + O_2$ | $1.15 \times 10^{-12}$ | 430 | (Saunders et al., 2003) |
| $HOOCH_2S + NO_2 \rightarrow HOOCH_2SO + NO$ | $6.00 \times 10^{-11}$ | 240 | (Saunders et al., 2003) |
| $HOOCH_2SO + O_3 \rightarrow SO_2 + CH_2O + OH + O_2$ | $4.00 \times 10^{-13}$ | | (Saunders et al., 2003) |
| $HOOCH_2SO + NO_2 \rightarrow SO_2 + CH_2O + OH + NO$ | $1.20 \times 10^{-11}$ | | (Saunders et al., 2003) |

[a] $2.24 \times 10^{11} \exp(-9.8 \times 10^3 / T) \exp(1.03 \times 10^8 / T^3)$

[b] $6.09 \times 10^{11} \exp(-9.5 \times 10^3 / T) \exp(1.1 \times 10^8 / T^3)$

where $T$ is air temperature.

### 235 2.3.2 Gas-phase reactions of the OH-addition Pathway

**Table 6** summarizes the new gas-phase reactions in the OH-addition pathway of DMS oxidation. We update the gas-phase reactions in the model to consider the oxidation of DMS by not only OH and NO₃ but also BrO, O₃, and Cl as recommended or reported in previous studies, e.g., Barnes et al. (2006), Hoffmann et al. (2016), and Chen et al. (2018). The new reactions producing dimethyl sulfoxide

(DMSO: $CH_3SCH_3O$), methanesulfinic acid (MSIA: $CH_3SOOH$), and MSA intermediates, are added to the model as new advected chemical tracers which undergo not only chemical production and loss but also transport and deposition. MSA and $SO_2$ are terminating products of these new gas-phase OH-addition pathway reactions, which is consistent with various modeling studies, e.g., Pham et al. (1995), Spracklen et al. (2005), and Chen et al. (2018). All oxidants (OH, $O_3$, $H_2O_2$, BrO, and HOBr) are

simulated online by the standard gas-phase chemistry scheme of CAM6-chem.



**Table 6.** Gas-phase DMS oxidation (OH-addition pathway) implemented into CAM6-chem in this study.

| Gas-phase Reactions | $k_{298}$ (cm$^3$ molecule$^{-1}$ s$^{-1}$) | $-E_a/R$ (K) | References |
|---|---|---|---|
| DMS + OH → 0.6SO$_2$ + 0.4DMSO + CH$_3$O$_2$ | See note[a] | - | (Burkholder et al., 2015; Pham et al., 1995) |
| DMS + NO$_3$ → SO$_2$ + HNO$_3$ + CH$_3$O$_2$ + CH$_2$O | $1.13 \times 10^{-12}$ | 530 | (Burkholder et al., 2015) |
| DMS + BrO → DMSO + Br | $3.39 \times 10^{-13}$ | 950 | (Burkholder et al., 2015) |
| DMS + O$_3$ → SO$_2$ | $1.00 \times 10^{-19}$ | 0 | (Burkholder et al., 2015) |
| DMS + Cl → 0.5SO$_2$ + 0.5DMSO + 0.5HCl + 0.5ClO | $3.40 \times 10^{-10}$ | 0 | (Burkholder et al., 2015) |
| DMSO + OH → 0.95MSIA + 0.05SO$_2$ | $8.94 \times 10^{-11}$ | 800 | (Burkholder et al., 2015) |
| MSIA + OH → 0.9SO$_2$ + 0.1MSA | $9.00 \times 10^{-11}$ | 0 | (Burkholder et al., 2015) |
| MSIA + O$_3$ → MSA | $2.00 \times 10^{-18}$ | 0 | (Lucas, 2002) |

[a] $k(T, [O_2], [M]) = 8.2 \times 10^{-39}[O_2]e^{(5376/T)} / (1 + 1.05 \times 10^{-5}([O_2]/[M])e^{(3644/T)})$ (cm$^3$ molecule$^{-1}$ s$^{-1}$)

### 2.3.3 Aqueous-phase reactions of the OH-addition Pathway

We also introduce new aqueous-phase reactions of the OH-addition pathway as shown in **Table**
**7.**

**Table 7.** Aqueous-phase DMS oxidation (OH-addition pathway) implemented into CAM6-chem in this study.

| Aqueous-phase Reactions | $k_{298}$ (M$^{-1}$ s$^{-1}$) | $-E_a/R$ (K) | References |
|---|---|---|---|
| DMS $_{(aq)}$ + O$_3$ $_{(aq)}$ → DMSO $_{(aq)}$ + O$_2$ $_{(aq)}$ | $8.61 \times 10^8$ | $-2600$ | (Gershenzon et al., 2001) |
| DMSO $_{(aq)}$ + OH $_{(aq)}$ → MSIA $_{(aq)}$ | $6.63 \times 10^9$ | $-1270$ | (Zhu et al., 2003) |



| | | | |
|---|---|---|---|
| MSIA $_{(aq)}$ + OH $_{(aq)}$ → MSA $_{(aq)}$ | $6.00 \times 10^9$ | 0 | (Sehested and Holcman, 1996) |
| MSI$^-$ $_{(aq)}$ + OH $_{(aq)}$ → MSA $_{(aq)}$ | $1.20 \times 10^{10}$ | 0 | (Bardouki et al., 2003) |
| MSIA $_{(aq)}$ + O$_3$ $_{(aq)}$ → MSA $_{(aq)}$ | $3.50 \times 10^7$ | 0 | (Hoffmann et al., 2016) |
| MSI$^-$ $_{(aq)}$ + O$_3$ $_{(aq)}$ → MSA $_{(aq)}$ | $2.00 \times 10^6$ | 0 | (Flyunt et al., 2001) |
| MSA $_{(aq)}$ + OH $_{(aq)}$ → SO$_4^{2-}$ $_{(aq)}$ | $1.50 \times 10^7$ | 0 | (Hoffmann et al., 2016) |
| MS$^-_{(aq)}$ + OH $_{(aq)}$ → SO$_4^{2-}$ $_{(aq)}$ | $1.29 \times 10^7$ | −2630 | (Zhu et al., 2003) |

Following a similar treatment employed by the Community Multiscale Air Quality (CMAQ)
Model version 5.1 (Fahey et al., 2017), we calculate, for each species involved in the new aqueous-
phase reactions, the phase transfer equations for gas-aqueous partitioning as below:

$$\frac{dC_g}{dt} = -k_{(g)\rightarrow(a)}C_g + k_{(a)\rightarrow(g)}C_a = -(k_t \, \text{LWC}) \, C_g + \left(\frac{k_t}{HRT}\right)C_a \qquad (3)$$

$$\frac{dC_a}{dt} = k_{(g)\rightarrow(a)}C_g - k_{(a)\rightarrow(g)}C_a = (k_t \, \text{LWC}) \, C_g - \left(\frac{k_t}{HRT}\right)C_a \qquad (4)$$

$$k_t = \left(\frac{r^2}{3D_g} + \frac{4r}{3c\alpha}\right)^{-1} \qquad (5)$$

where $C_g$ and $C_a$ are the gas-phase and aqueous-phase concentration of a species involving in reactions
in **Table 7**; $k_{(g)\rightarrow(a)}$, $k_{(a)\rightarrow(g)}$ (s$^{-1}$) are its gas-to-aqueous and aqueous-to-gas phase-transfer coefficients;
$k_t$ (s$^{-1}$) is its base phase-transfer coefficient; $H$ (M atm$^{-1}$) is its effective Henry's Law constant; $r$ (cm)
is its mean particle radius; $D_g$ is gas phase diffusion coefficient (assumed at 0.1 cm$^2$ s$^{-1}$ here following
Dentener and Crutzen (1993)); $c$ (cm s$^{-1}$) is its thermal speed, and; $\alpha$ is its mass accommodation
coefficient. Values of $H$ and $\alpha$ for DMS, DMSO, MSIA, and MSA are given in **Table 8**. LWC (cm$^3$-
water (cm$^3$-air)$^{-1}$) is liquid water content of interstitial aerosol determined by MOSAIC (Zaveri et al.,
2021) or cloud liquid water content calculated by the CAM6; $R=0.082$ (L atm K$^{-1}$ mol$^{-1}$) is the universal
gas constant, and; $T$ (K) is air temperature.





**Table 8.** Summary of parameters of DMS and its oxidation intermediates used in this study.

| | $H_{298}$ (Henry's Law Coefficients at 298K) [M/atm] [a, b] | $-E/R$ (Heat of dissolution for H298)/R [K] [a, b] | $K_1$ (acid dissociation constant) [M] [b] | $\alpha$ (mass accommodation coefficient) [c] | molar mass [g/mol] [a] |
|---|---|---|---|---|---|
| **DMS** | 0.54 | 3460 | - | 0.001 | 62.1324 |
| **DMSO** | $10^7$ | 2580 | - | 0.1 | 78.13 |
| **MSIA** | $10^8$ | 1760 | −2.28 | 0.1 | 80.11 |
| **MSA** | $10^9$ | 1760 | 1.86 | 0.1 | 96.1 |

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

## 3 Implications of the Extended DMS Oxidation Mechanism

### 3.1 Global sulfur budget and distribution in present-day

The global burden of DMS in [MOD_2000] is 50 Gg-S. It is 38% lower than the standard run, [STD_2000], but remains within the range of 9.6–140 Gg-S from other studies (Faloona, 2009; Kloster et al., 2006). **Figure 2** shows the reduction is mainly over the Southern Ocean and is attributable to faster chemical losses via DMS+BrO and DMS$_{(aq)}$+O$_{3(aq)}$ (**Figure 3**). The global lifetime of DMS decreases from 1.5 days in [STD_2000] to 0.8 days in [MOD_2000]. These values are comparable to the range of 1.2–2.1 days reported in Chen et al., (2018).



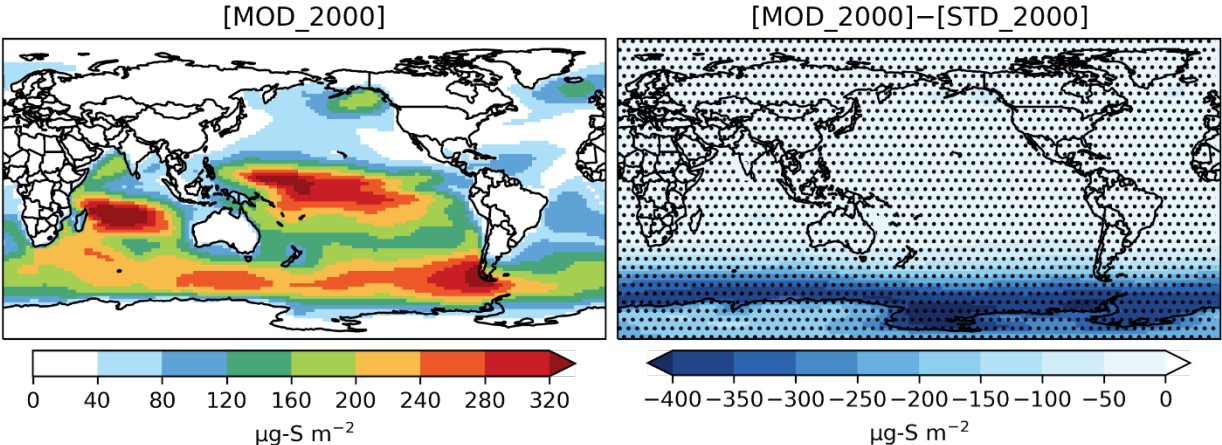

**Figure 2.** Spatial distribution of annual-mean column concentration (μg-S m$^{-2}$) for DMS simulated by [MOD_2000] (left), and its difference from the baseline run, i.e. [MOD_2000]–[STD_2000] (right). Dotted regions (nearly worldwide) indicate where statistically significant differences are identified by grid-by-grid two-sample t-tests with $p$-values < 0.05.

Globally, chemical loss is the largest sink of DMS (~24 Tg-S yr$^{-1}$) in both PD simulations. The model default chemistry in [STD_2000] predicts that OH oxidation makes up 40% (H-abstraction) and 39% (OH-addition) of DMS chemical removal globally while the remaining portion is attributed to NO$_3$ oxidation. These three reactions are only responsible for 80% of global DMS loss in [MOD_2000]. **Figure 3** shows that, in [MOD_2000], DMS is mainly oxidized by OH in the gas phase (34% via the H-abstraction channel and 23% via the OH-addition pathway) which contributes up to 80% of local loss over the tropical oceans, where surface OH is the highest. The annual-mean surface concentrations of all oxidants which react with DMS in our updated scheme are summarized in **Figure S1**. NO$_3$ oxidation of DMS accounts for 23% of global DMS chemical loss and is dominant in Northern Hemisphere mid-latitudes, where the outflow of nitrogen oxides (NO$_x$) – precursors of atmospheric NO$_3$ – from land are substantial (Miyazaki et al., 2012). DMS oxidation by NO$_3$ contributes <10% over most marine environments in the Southern Hemisphere. Previous studies estimate that the global





contribution of OH and NO$_3$ to DMS oxidation range ~50–70% and 15–30%, respectively (Berglen, 2004; Boucher et al., 2003; Khan et al., 2016; Chen et al., 2018).

Oxidation by BrO is responsible for 11% of global DMS removal, which falls midway within
the previously estimated range of 8–29% (Boucher et al., 2003; Khan et al., 2016; Chen et al., 2018). Regionally, its importance can be up to 50–60% over the high latitudes in the Southern Hemisphere, which is close to a previous box model experiment (Hoffmann et al., 2016).

DMS+O$_3$ is the only multi-phase DMS oxidation reaction in this study, accounting for 8% (aqueous-phase) and 0.4% (gas-phase) of global DMS depletion. The oxidation rates via these reactions
estimated by Boucher et al. (2003) were 6% and 3% of the total DMS sink calculated, respectively. Our lower gas-phase DMS+O$_3$ reaction rate could be due to the inclusion of the BrO oxidation, which is missing in their study. Regionally, the fractional contribution of aqueous-phase DMS+O$_3$ to DMS oxidation can be up to 20–30% over high-latitude oceans, which is on the upper end of the range of 5–30% high-latitude DMS losses previously reported (von Glasow and Crutzen, 2004; Chen et al., 2018).

Lastly, the Cl oxidation reactions via either the addition or abstraction channels contribute equally (0.3% each, globally) to the chemical removal of DMS, which is consistent with the proposal of (Atkinson et al., 2004). Our estimated values are much lower than the 4% found in a global model study (Chen et al., 2018) and the 8–18% from box-model studies (von Glasow and Crutzen, 2004; Hoffmann et al., 2016).






**Figure 3.** Spatial distribution of fractional DMS oxidation (%) from [MOD_2000] through DMS+OH (abstraction), DMS+OH (addition), DMS+NO$_3$, DMS+BrO, DMS$_{(aq)}$+O$_{3(aq)}$, DMS+O$_3$, DMS+Cl (abstraction), and DMS+Cl (addition). Percentages in the brackets denote contribution to global chemical loss. Subplots are arranged in descending order of their annual-total oxidation rates.





The global atmospheric sulfur burden is increased by 41 Gg-S (or 4.1%) from [STD_2000] to [MOD_2000] (**Figure 4** and **Table S1**). Approximately half (23 Gg-S) of this increment is associated

with the recovery of the missing sulfur associated with the OH-addition reaction in the standard chemistry (the second reaction in **Table 3**), which does not conserve sulfur. The remaining total sulfur burden increase is attributable to the extended chemistry scheme. As discussed above, the DMS burden in [MOD_2000] is lower than [STD_2000] by 38% due to faster oxidation. This oxidation produces intermediates with a wide range of lifetimes. The addition of intermediates with relatively long physical

lifetimes (to dry and wet deposition only) of HPMTF (1,300 days) and MSA (8.5 days) delays the formation of $SO_2$ (2.6 days) and sulfate (4.4 days) compared to the standard reactions in [STD_2000], which increases the export of sulfur-containing intermediates.

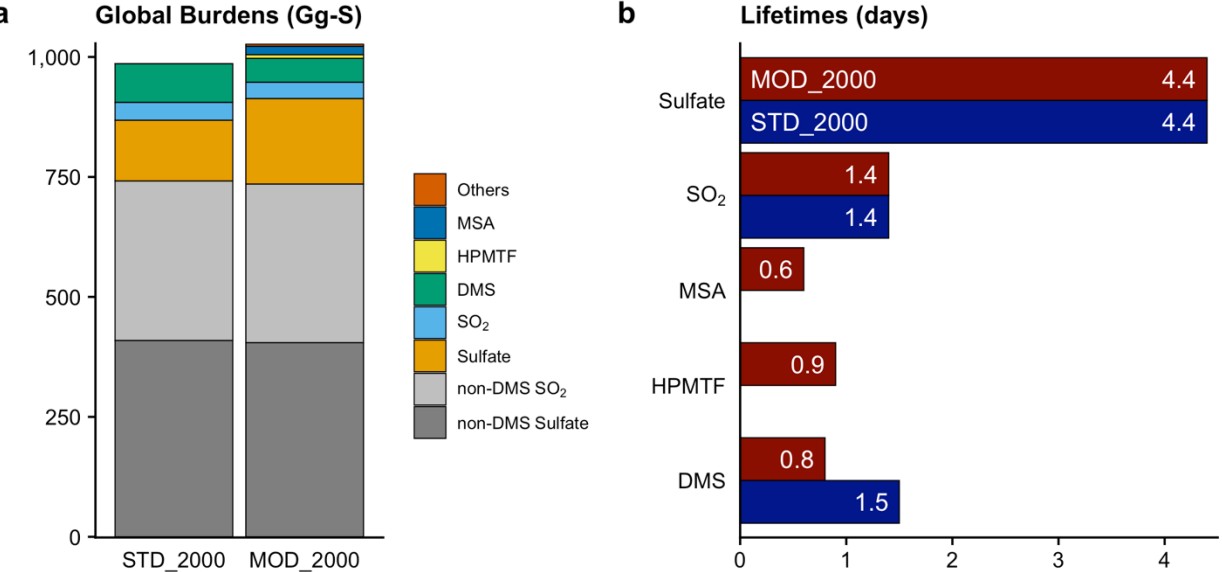



**Figure 4. (a)** Global burdens of various atmospheric sulfur species in our simulations. "Others" includes all other sulfur-containing intermediates in the new chemistry, e.g., DMSO, MSIA, etc. SO$_2$ (blue) and sulfate (orange) refer to the burden of these species that originate from DMS oxidation only; non-DMS contributions are shown in grey. **(b)** Total lifetimes of the atmospheric sulfur species to both physical and chemical losses.


The PD global annual mean burden for sulfate aerosol is 582 Gg-S in [MOD_2000], with an interannual variability of 46 Gg-S (standard deviation of annual means). It is comparable to the 580 Gg-S in a previous CAM6-chem study (Tilmes et al., 2019) and is within the estimates (420–660 Gg-S) from studies using other models (Heald et al., 2014; Chen et al., 2018). The new DMS chemistry has

increased the global sulfate burden by 47 Gg-S (or 8.8%) from the baseline value of 535 Gg-S in [STD_2000]. The statistically significant increases in sulfate resulting from the expanded chemistry are mostly found over the tropical and sub-tropical oceans in the Southern Hemisphere (**Figure 5**). There is no strong seasonality in the additional sulfate produced from our expanded chemistry. We estimate that the sulfate burden attributable to DMS increases by 41% from 126 Gg-S in [STD_2000] to 178

Gg-S in [MOD_2000]. Most of this increase in sulfate burden (72%) comes from the expansion of the gas-phase chemistry with a minor additional contribution from the aqueous-phase chemistry. In a sensitivity test where the isomerization branch reaction (**Table 5**) is removed from [MOD_2000], the global DMS-derived sulfate burden is reduced by 2.0% (relative to [MOD_2000]).


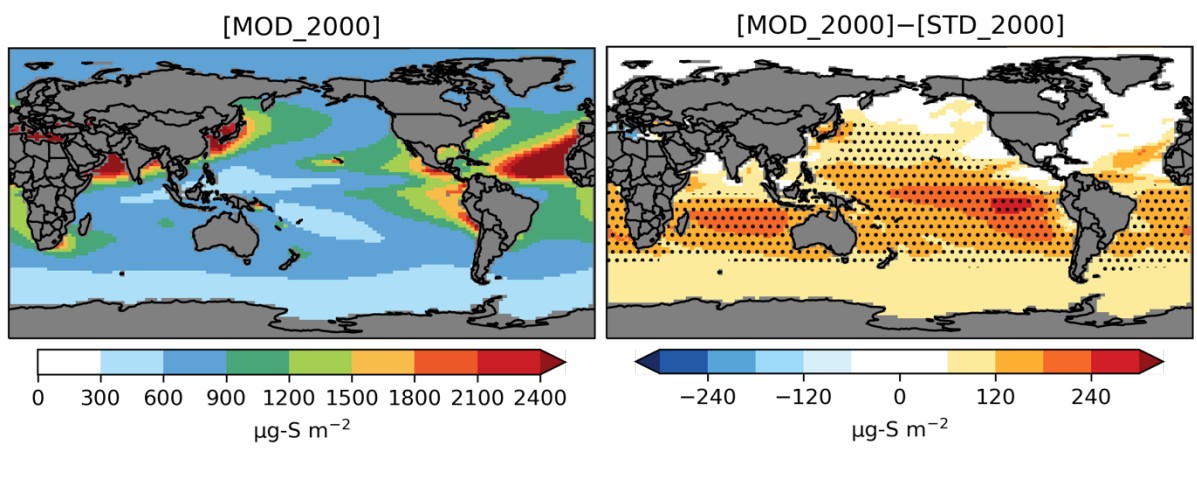


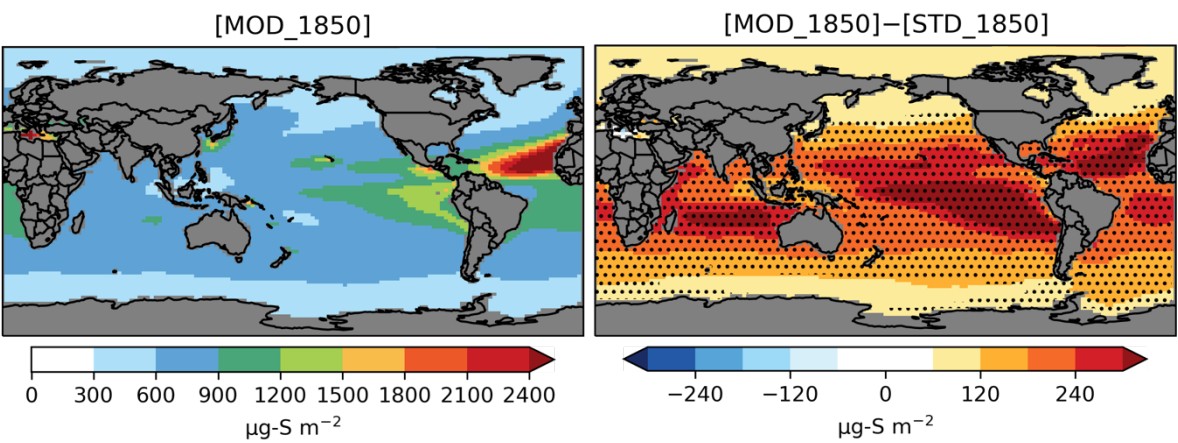

**Figure 5.** Spatial distribution of annual-mean column concentrations (μg-S m$^{-2}$) for sulfate aerosol simulated by [MOD] (left column), and their difference from the baseline run (right column). Only values over the ocean are shown. Dotted regions indicate where statistically significant differences are identified by grid-by-grid two-sample t-tests with $p$-values $< 0.05$.




The spatial distribution of the product branching ratios of DMS oxidation is shown in **Figure 6**. In addition to depositional removal, HPMTF converts into $SO_2$ while $SO_2$ and MSA are then oxidized to form sulfate. We estimate that 33% of the annual-total DMS oxidation will yield HPMTF. This is comparable to the observationally constrained estimates from ATom-3 and ATom-4 flight campaigns, that ~30–40% DMS was oxidized to HPMTF along their flight tracks (Veres et al., 2020). High HPMTF production is typically seen in the summer MBL, coinciding with the HPMTF hotspots over tropical oceans as shown in **Figure S2**. To address the uncertainty in the production and loss of HPMTF as discussed in **Sect. 2.3**, we run several sensitivity tests using five combinations of $k_{iso}$ and $k_{HPMTF+cloud}$ values based on [MOD_2000]. **Table S2** and **Figure S5** summarize the key results of these sensitivity tests. Compared to [MOD_2000], we find that using a faster $k_{iso}$ of 0.12 $s^{-1}$ at 293 K (Ye et al., 2021) increases the global annual-total isomerization rate of MSP by 5.6% while the global burden of HPMTF increases by 4.1%. Increasing the isomerization rate has little impact on the burden of sulfate from DMS (increase of only 4.0%). We also evaluate the importance of cloud uptake of HPMTF with two hypothetical values of $k_{HPMTF+cloud}$ at $5 \times 10^{-3}$ $s^{-1}$ (Vermeuel et al., 2020) and $5 \times 10^{-5}$ $s^{-1}$. At these rates, cloud uptake becomes an important sink of HPMTF, responsible for 68–69% and 28% of total HPMTF losses respectively. The corresponding global burdens of HPMTF are substantially lowered by 85–86% and 52%. For simulation with cloud uptake loss, the burdens of HPMTF and sulfate are much less sensitive to our choice of $k_{iso}$ due to the rapid loss of HPMTF to cloud uptake. In these sensitivity simulations, the sulfur contained in HPTMF is assumed to be removed from the system once taken up by cloud, thus reducing the sequential formation of $SO_2$ and sulfate (by up to 8 %).





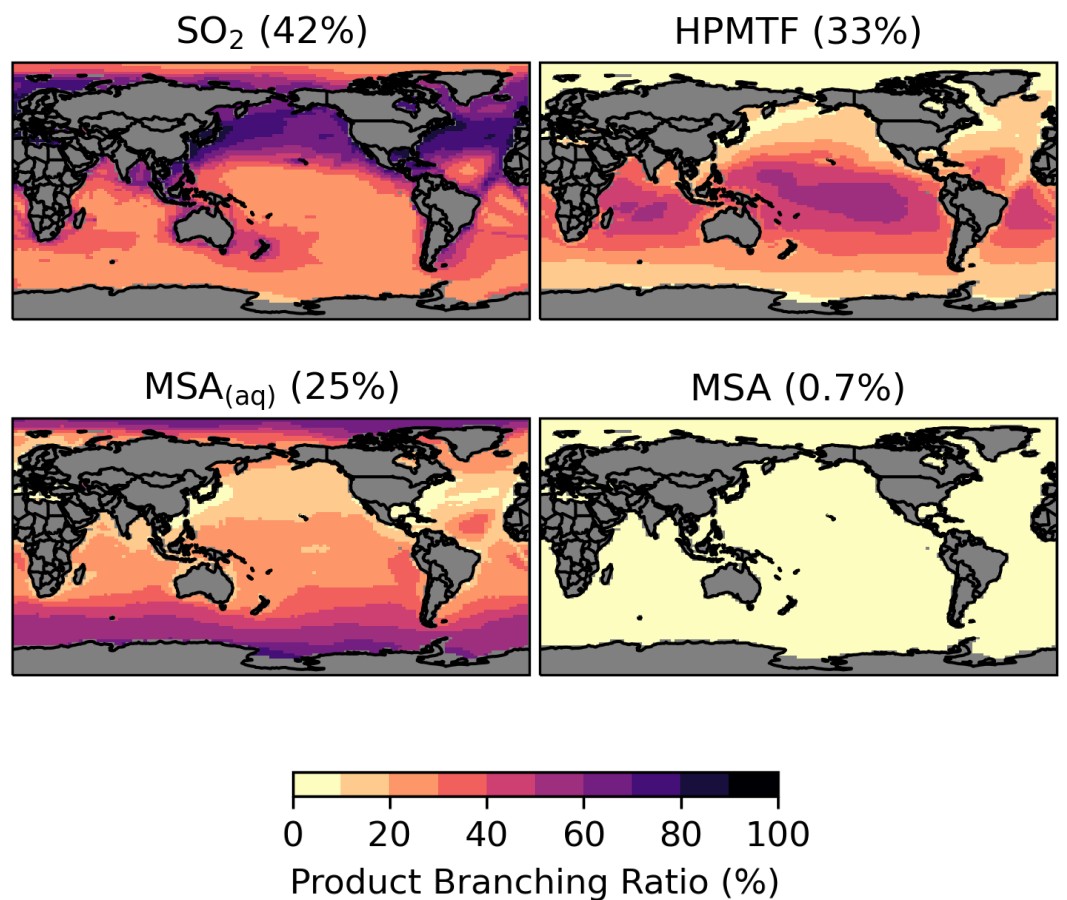

**Figure 6.** Branching ratio (%) of the multi-phase DMS oxidation pathways in [MOD_2000], considering HPMTF, $SO_2$, and MSA as terminating products estimated from their annual-total production rates.

MSA is a key intermediate generated from the OH-addition channel of the multi-phase DMS oxidation, especially, over remote marine atmosphere. Our result shows that aqueous-phase MSA formation accounts for most of MSA production as commonly reported (von Glasow and Crutzen, 2004; Barnes et al., 2006; Zhu et al., 2006; Hoffmann et al., 2016; Chen et al., 2018). In [MOD_2000], the global MSA burden is 7.5 Gg-S, which is smaller than the range of 13–40 Gg-S from previous model



studies (Pham et al., 1995; Chin et al., 1996, 2000; Cosme et al., 2002; Hezel et al., 2011; Chen et al.,
2018). In [MOD_2000]. Most MSA is formed over the Southern Ocean (**Figure S3**). The lifetime of
MSA is 0.6 days globally, shorter than the 5–7 days previously proposed (Chin et al., 1996, 2000;
Cosme et al., 2002; Hezel et al., 2011; Pham et al., 1995), likely because we include the aqueous-phase
OH-oxidation to sulfate which is a significant loss process for MSA. This oxidation accounts for ~76%

of removal in [MOD_2000], followed by cloud uptake (18%).

### 3.2 Comparison with observations

**Table 9** summarizes the key observational datasets used here to compare with our PD model
simulations for their wide coverage of the remote marine atmosphere. The VAMOS Ocean-Cloud-

Atmosphere-Land Study Regional Experiment (VOCALS-REx) is an international field project that
took place during October and November in 2008 over southeastern Pacific off northern Chile and
southern Peru (Wood et al., 2011). VOCAL-REx consists of both ship-based and airborne
measurements for lower-atmospheric DMS, $MSA_{(aq)}$, and non-sea-salt sulfate aerosol (nss-$SO_4^{2-}{}_{(aq)}$).
The Atmospheric Tomography (ATom) mission of NASA is a flight campaign spanning from the Arctic

to the Antarctic over the remote Pacific and Atlantic Oceans between 2016 and 2018, (Wofsy et al.,
2018). During ATom, an array of instruments was used to collect and analyze daytime air samples from
the remote marine environments, providing measurements of DMS, HPMTF, $MSA_{(aq)}$, and nss-$SO_4^{2-}{}_{(aq)}$. The Aerosol and Cloud Experiments in the Eastern North Atlantic (ACE-ENA) probed the
atmosphere surrounding the ENA observatory on Graciosa Island during summer 2017 and winter 2018

(Wang et al., 2019a, 2021). ACE-ENA provides high time-resolution in-situ measurements of MSA
and sulfate aerosol in the lower troposphere (Zawadowicz et al., 2021). We note that the model-
measurement comparisons are not exact, given that our simulations are performed using free-running
dynamics and thus are not paired to the meteorological year of measurements. We therefore sample
monthly mean values from the model at the location of the observations to provide qualitative

comparisons.





**Table 9.** Key observational datasets used in this study.

| Mission | Aliases | Instruments | Species Measured | Reference |
|---|---|---|---|---|
| VOCALS-REx | OBS_BROWN | Quadrupole mass spectrometry on Research Vessel (R/V) Ronald H. Brown | DMS | (Huebert et al., 2004) |
| | OBS_C130 | Quadrupole mass spectrometry on NSF/NCAR Lockheed C-130 aircraft | DMS | (Booth and Powell, 2006) |
| | OBS_PILS | Particle into liquid sampler (PILS) on Department of Energy (DoE) Gulfstream-1 (G-1) aircraft | $MSA_{(aq)}$, $nss\text{-}SO_4^{2-}{}_{(aq)}$ | (Allen et al., 2011) |
| | OBS_AMS | Aerosol mass spectroscopy (AMS) and two-stage impactor (TSI) on R/V Ronald H. Brown | $MSA_{(aq)}$, $nss\text{-}SO_4^{2-}{}_{(aq)}$ | (Huebert et al., 2004) |
| ATom | OBS_WAS | Whole Air Sampling (WAS) | DMS | (Simpson et al., 2001) |
| | OBS_TOGA | Trace Organic Gas Analyzer (TOGA) | DMS | (Apel et al., 2015) |
| | OBS_CIMS | Iodide-ion chemical ionization time-of-flight mass spectrometer (CIMS) | HPMTF | (Veres et al., 2020) |
| | OBS_AMS | high-resolution time-of-flight mass spectrometer (HR-ToF-AMS) | $MSA_{(aq)}$, $nss\text{-}SO_4^{2-}{}_{(aq)}$ | (Canagaratna et al., 2007) |
| ACE-ENA | OBS_AMS | HR-ToF-AMS | $MSA_{(aq)}$, $nss\text{-}SO_4^{2-}{}_{(aq)}$ | (Zawadowicz et al., 2021) |

Most DMS resides in the lower troposphere (**Figure 7**). Annual-mean surface DMS from [MOD_2000] ranges from 40-300 ppt over much of the ocean, but can exceed 320 ppt over the Southern Ocean and northeastern Pacific, regions with high DMS emissions. DMS concentrations of ~25–125 ppt were observed at Cape Grim, Australia in 1990-1993 (Ayers et al., 1995). Sciare et al. (2000) report an annual-mean DMS of 181 ppt at Amsterdam Island in the Indian Ocean during the 1990s. Both values are in line with the surface DMS at the corresponding locations modeled by [MOD_2000].

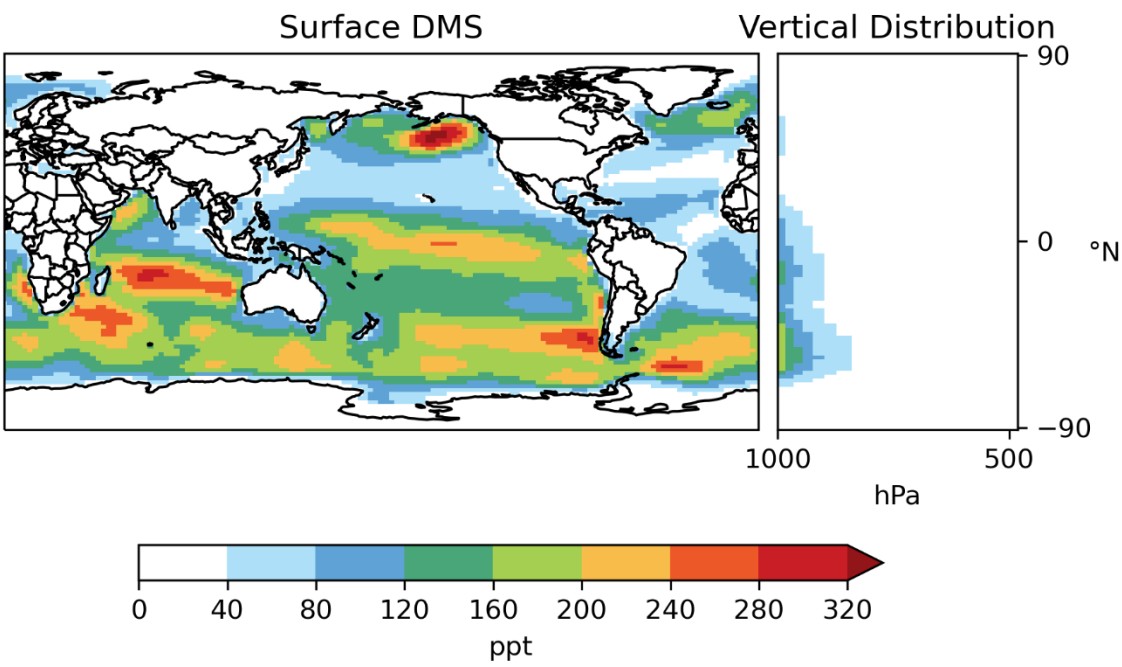

**Figure 7.** Horizontal distribution of annual-mean surface mixing ratio and zonal-mean vertical distribution of DMS (both in ppt) modeled by [MOD_2000].

**Figure 8** summarizes the spatial difference between the observed DMS from the VOCALS-REx and ATom missions and the simulated DMS. The model captures the peaks over the tropical Pacific and the Southern Oceans off the coast of South America, but aircraft measurements detect hotspots that are not simulated by the model (**Figure 8(a)**). During VOCALS-REx the ship-based measurements (BROWN) recorded a range of near-surface DMS from 18 ppt to 111 ppt, while the airborne measurements (C130) reveal a vertically decreasing trend of DMS mixing ratios, from 33 ppt at ~500 m to 2.0 ppt at ~2 km (**Figure 8(b)**). Modeled surface DMS falls in the range of the ship measurements. Compared to the aircraft observations, simulated DMS is biased high at the surface and declines more abruptly, which may indicate biases in vertical mixing or cloud processing. DMS concentrations are slightly lower in the simulation with updated chemistry [MOD_2000], but follow



the same vertical profile. We disaggregate the ATom observations into three regional groups, namely

Pacific, Atlantic, and Southern Oceans as in **Figure 8(a)**. DMS concentrations were measured by two

instruments during ATom (WAS and TOGA, the former generally reported higher values); both are

compared with model values in **Figure 8(c)**. Observed DMS concentrations during ATom are

substantially lower than measured during VOCALS, and lower than any region simulated by the model.

Modeled DMS is biased high in all three regions, especially over the Southern region where the

discrepancy extends up to 5 km. The new chemistry increases DMS losses and shortens the DMS

lifetime, reducing the model bias in [MOD_2000]. The decrease in simulated DMS is largest over the

Southern Oceans (–49% at the surface), where oxidation by BrO and $O_3$ in the aqueous phase are

important and the model-observation bias is substantially reduced. The remaining model biases during

ATom exceed the uncertainty of the kinetics for the current DMS oxidation scheme and are likely

attributable, at least in part, to the uncertainty in DMS emission. A sensitivity test where we reduce the

sea surface DMS concentration by 50% in regions south of 30ºS in [MOD_2000] produces, as expected,

a comparable decrease in DMS mixing ratios in the lower atmosphere (<5 km), and the model-

observation deviations are further narrowed (see **Figure S4** and **Supplementary Information** for

details). Constraining DMS emissions is beyond the scope of this work, but is clearly a major source of

uncertainty.







**Figure 8. (a)** Measurements of DMS during VOCALS-REx (diamonds in southeast Pacific near the coastline of Peru) and ATom (dots) missions. Measured values showing are local 90-percentiles above oceans. ATom data are grouped into three regions as shown in the purple dashed polygons. Marine-only annual-mean near-surface (>500 hPa) DMS mixing ratios from [MOD_2000] are shown in the background as a reference. Vertically-binned modeled and observed medians of DMS, during **(b)** VOCALS-REx and **(c)** ATom, are shown. Error bars and gray shadings indicate data ranged between corresponding upper and lower quantiles.


**Figure 9** compares the mean vertical profile of HPMTF mixing ratios observed during ATom against the model [MOD_2000]. Over the Pacific and Atlantic regions, HPMTF mixing ratios are largest at lower altitudes and decrease to <1 ppt in the middle and upper troposphere. The model generally reproduces the observed magnitude and vertical profile. The model [MOD_2000] is biased high over the Southern Ocean region, particularly in the lower troposphere. Such high biases are consistent with the aforementioned overestimation of DMS over this region. In the lower atmosphere over tropical and mid-latitude oceans, the modeled DMS:HPMTF ratios range from 5:1 to 2:1, which is larger than the 1:1 ratio observed during ATom (Veres et al., 2020), suggesting that the model may underestimate the DMS-to-HPMTF conversion rate or overestimate the HPMTF loss. Our model predicts that OH-oxidation to $SO_2$ is dominant in the removal of HPMTF while dry and wet deposition are negligible. The addition of cloud uptake (discussed in **Sect. 2.3.2**) can dramatically decrease HPMTF concentration (by up to >73% when assuming moderate cloud uptake rate of $5 \times 10^{-5}$ $s^{-1}$), resulting in a better model-observation agreement in the lower troposphere over the Southern Ocean but low biases over the Pacific and Atlantic (**Figure 9**). In light of the DMS biases of **Figure 8**, this irreversible cloud uptake may over-correct the HPMTF concentrations.





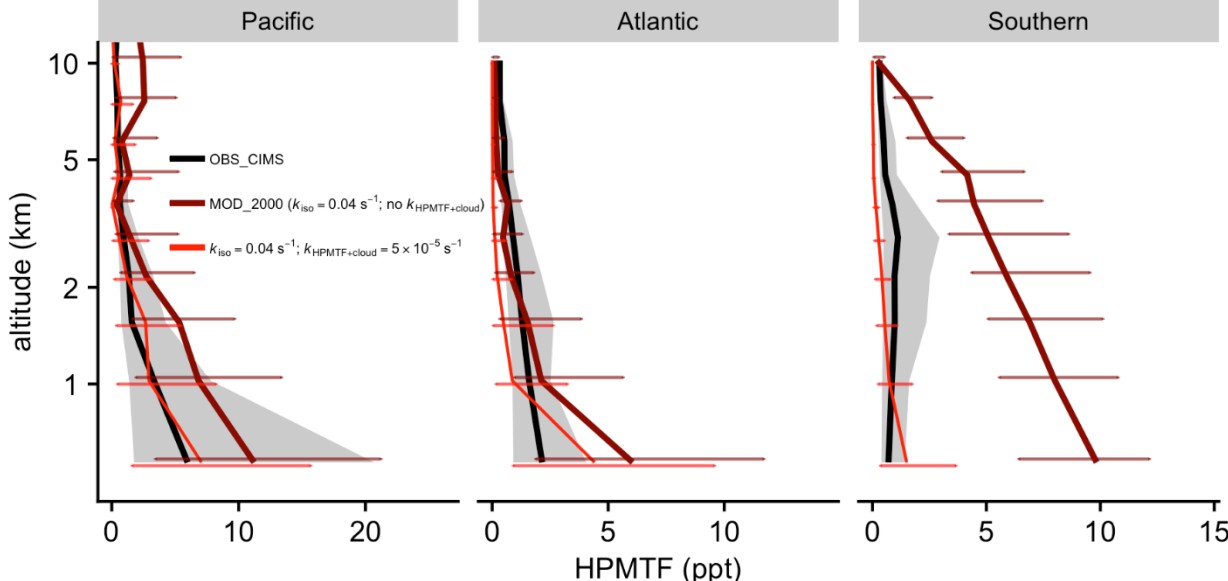

**Figure 9.** Measured (ATom) and modeled values of HPMTF, vertically binned. The thick lines show medians. Error bars and gray shadings indicate data ranged between corresponding upper and lower quantiles. The thin red line indicates the results from a sensitivity test with $k_{iso}$ at 0.04 s$^{-1}$ and $k_{HPMTF+cloud}$ at $5 \times 10^{-5}$ s$^{-1}$.

Our simulation shows that the gas-phase MSA formation is small compared to aqueous-phase formation, in line with previous work (Barnes et al., 2006; von Glasow and Crutzen, 2004; Zhu et al., 2006; Hoffmann et al., 2016; Chen et al., 2018; Hoffmann et al., 2021). Near the sea surface, simulated gas-phase MSA is < 0.03 ppt even in Southern Hemisphere summer, while a recent ship-based measurement reported an averaged concentration ranged 1.4–25 ppt (Yan et al., 2019). The model also substantially underestimates gas-phase MSA (<0.001 ppt) when compared to a wintertime site measurements in Germany (0.5–10 ppt) (Stieger et al., 2021). **Figure 10** shows the concentration of submicron particulate MSA measured during ATom and the collocated concentration of MSA aerosol in Aitken and accumulated modes modeled by [MOD_2000]. The model overestimates the mid-



tropospheric MSA concentrations in the Southern Ocean during ATom. Conversely, over the Pacific

and Atlantic, the model underestimates MSA at mid- and low altitudes. Over the southeastern Pacific,

the measured submicron MSA from VOCALS-REx ranged from 50–80 ng m$^{-3}$ at lower altitudes (<1

km) (Wood et al., 2011) while the location-matched simulated MSA was considerably lower (6–15 ng

m$^{-3}$). Above Graciosa Island in the north Atlantic, ACE-ENA-observed MSA ranged from 10–20 ng

m$^{-3}$ at the lower troposphere (<1 km) and gradually reduced to ~5 ng m$^{-3}$ in the mid-troposphere (2–5

km) (Zawadowicz et al., 2021), whereas the model estimates a negligible amount of MSA (<0.5 ng m$^{-3}$) near the Azores, as a result of limited DMS emissions in that region (as reflected by low DMS

concentrations in **Figure 2**). Outside of the Southern Ocean, the mean simulated concentration of MSA

is underestimated compared to all observations, which suggests that the MSA-forming branches of

DMS oxidation (H-abstraction where MSP reacts with NO, or multi-phase OH-addition reactions of

DMS) may be under-represented in our simulations, or that the loss of MSA (by reaction with OH , the

reaction rate for which is still highly uncertain (Milne et al., 1989; Zhu et al., 2006; Chen et al., 2018))

may be overestimated.

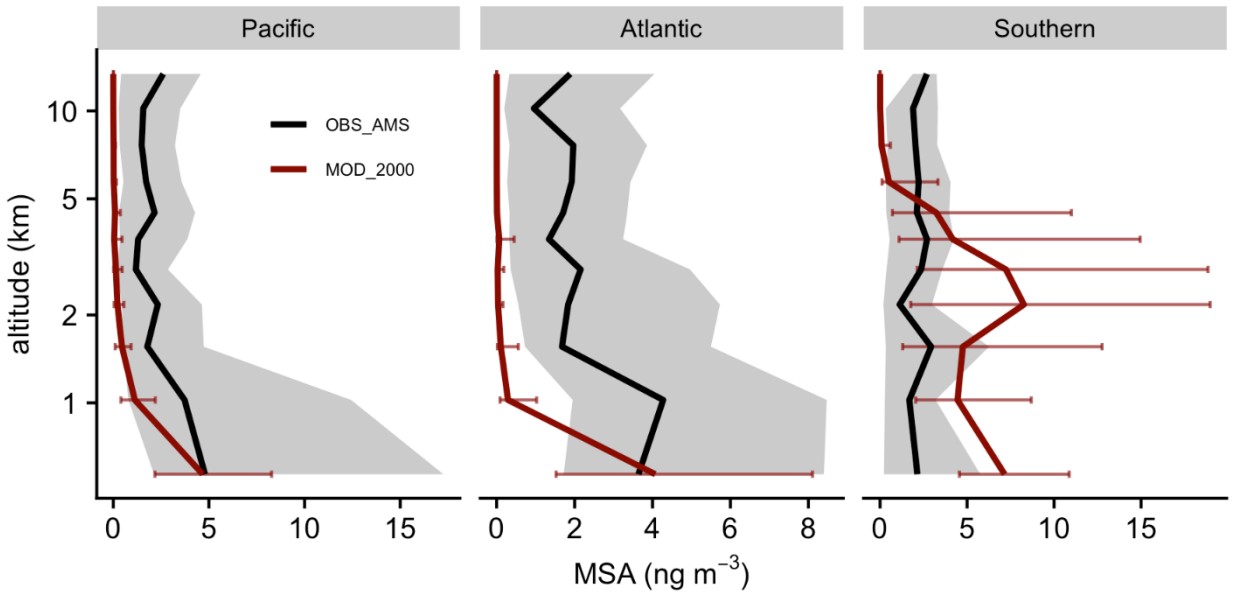





**Figure 10.** Medians of observed (ATom) and modeled concentration of MSA aerosol, vertically binned. The thick lines show medians. Error bars and gray shadings indicate data ranged between corresponding
upper and lower quantiles.

Concentrations of the sulfate aerosol simulated with both [STD_2000] and [MOD_2000] generally agree well with measurements from ATom (**Figure S6**). Our model also performs well at the surface when compared against VOCAL-REx and ACE-ENA, but is biased high above 1 km, likely
reflecting biases in anthropogenic sulfate exported from continental regions.

**3.3 Global sulfur budget and distribution in the pre-industrial era**

As seen under PD conditions, the formation of intermediates expands the overall lifetime of sulfur-containing species in the PI atmosphere, thereby increasing the natural sulfate aerosol
background. A summary of the burdens and lifetimes of the sulfur-containing species from the PI simulations is given in **Table S1**. The DMS burden in PI from [MOD_1850] is 84% larger than its PD counterpart due to slower oxidation which prolongs the atmospheric lifetime. Oxidation by OH via the H-abstraction (38% of total DMS oxidation in [MOD_1850]) and the OH-addition channels (27%) are still the primary loss pathways of DMS (**Figure S7**). DMS+NO$_3$ becomes less important (23% in PD
vs. 6.0% in PI) given reduced sources of NO$_x$, resulting in a lowered DMS-to-SO$_2$ conversion rate at 27%, compared to 47% in [MOD_2000] (**Figure S8**). Reduced NO$_x$ also limits the reaction rates of MSP+NO, thereby favoring the isomerization pathway (92% of total loss of MSP in PD vs. 96% in PI). Hence, the conversion of DMS to HPMTF becomes more important (39% of total DMS oxidation in PI), leading to a doubling of the HPMTF burden in the PI compared to PD. The addition pathway
producing MSA becomes more dominant over the tropical ocean via oxidation by OH and the high latitudes by DMS+BrO, raising the MSA burden by 59% compared to PD. Lastly, the expanded DMS oxidation chemistry increases the PI global annual-mean sulfate burden by 29% from 319 Gg-S





[STD_1850] to 412 Gg-S ([MOD_1850]) (**Figure 5**), of which 57% is derived from DMS, significantly larger than the 31% in PD, confirming that DMS is a relatively more important source of sulfate in PI.

Similar to PD, the majority (66%) of this additional sulfate in the PI is produced via the expanded gas-phase oxidation pathways and this addition is largely aseasonal. The absolute burden of sulfate produced from DMS oxidation is higher in the PI (236 Gg-S) compared to the PD (178 Gg-S).

## 4 Global Radiative Impacts of Updated DMS Chemistry

Changes to particle-phase sulfate and MSA due to the expanded DMS chemistry, as described above, may alter both aerosol-radiation and aerosol-cloud interactions. Given that particulate MSA is not included in the current CAM6-chem aerosol scheme, to account for its radiative impacts, we assume MSA interacts with radiation like sulfate optically by implementing an artificial rapid conversion of MSA to sulfate. These two adjusted cases are aliased as [MOD_RE_1850] and [MOD_RE_2000]

respectively. Details of this implementation are described in **Supplementary Information**.

### 4.1 Direct Radiative Effect (DRE)

Following the recommendation in Ghan (2013), we focus our analyses on the shortwave (SW) DRE. The PD global annual-mean sulfate DRE modeled with [MOD_RE_2000] and [STD_2000] are $-0.32$ W m$^{-2}$ and $-0.31$ W m$^{-2}$, respectively, which are slightly less negative than previous estimations

of $-0.36$ W m$^{-2}$ to $-0.42$ W m$^{-2}$ (Heald et al., 2014; Myhre et al., 2013; Yang et al., 2017). The global annual-mean cloud fraction in the model is 77% while the all-sky DRE is 59% of the estimate of clear-sky DRE. The more frequent cloudiness in CAM6-chem may explain the lower DRE compared to the AeroCom II models, which report a typical all-sky to clear-sky ratio of 1:2 (Myhre et al., 2013). The DMS-associated sulfate DRE in PD is $-0.11$ W m$^{-2}$ in [MOD_RE_2000], which is slightly stronger

than the value of $-0.074$ W m$^{-2}$ in a CAM5-chem study using the standard DMS oxidation chemistry (Yang et al., 2017) but substantially weaker than the $-0.23$ W m$^{-2}$ reported by Rap et al. (2013) using a different model. The DRE contribution of MSA alone is small ($-0.8$ mW m$^{-2}$; estimated by the DRE





difference of [MOD_RE_2000] minus [MOD_2000]). The sulfate DRE is not sensitive to HPMTF cloud loss given that this loss has a modest impact on the sulfate burden in our simulations. Our new

DMS chemistry has strengthened the PD sulfate direct cooling by 0.01 W m$^{-2}$ or 4% of the contribution attributed to DMS relative to [STD_2000].

The rise in sulfate burden from PI to PD driven by anthropogenic emissions occurs mainly over the land in the northern hemisphere; this impact is much larger than the increase in sulfate produced by the expanded DMS chemistry (**Figure 11)**. The zonal extrema of PD sulfate burden, aerosol optical

depth (AOD), and DRE are collocated around 30ºN (**Figure 11**). The larger difference in sulfate load in the southern tropics (30ºS to 0ºN) due to the new DMS oxidation reactions also translates to a larger DRE difference in those latitudes (**Figure S9**).

The direct radiative forcing (DRF) is estimated by differencing the DRE estimated with anthropogenic emissions at 1850- and 2000-levels. The DRF of [MOD_RE] and [STD] attributed to

sulfate and MSA aerosols are calculated as –0.11 W m$^{-2}$ and –0.13 W m$^{-2}$, respectively This difference indicates a relatively linear relationship between sulfate loading and DRF.

**a**

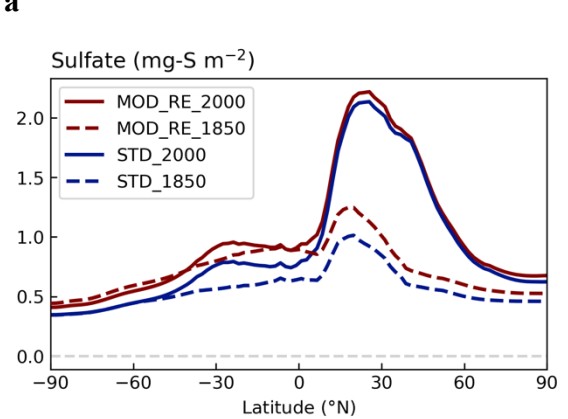

**b**

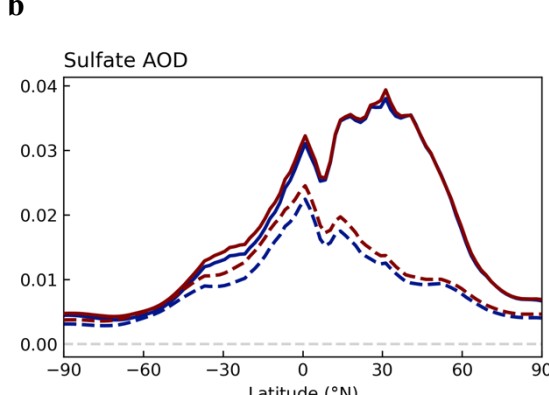



c

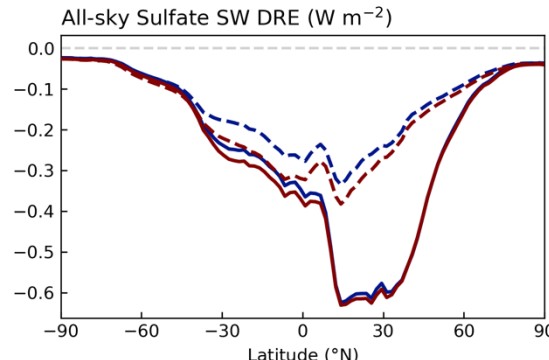

**Figure 11.** Contrasting the zonal-means of **(a)** sulfate column concentration, **(b)** sulfate AOD, and **(c)** all-sky SW sulfate DRE, modeled with [MOD_RE] and [STD] chemistry at the PI and PD emission levels. Note that particulate MSA is included as sulfate in the MOD_RE simulations.

## 4.2 Impacts on Aerosol-cloud Interactions and Indirect Radiative Forcing (IRF)

While the central estimate of the IRF of aerosols from the AR5, which reflects constraints from selected satellite and GCM analyses, is –0.45 W m$^{-2}$, the IRF estimated by the majority of models reported in AR5 (see Figure 7-19 of Boucher et al. (2013)) ranges from –1.0 to –2.5 W m$^{-2}$. This suggests that aerosol-cloud interactions may be substantially overestimated by the majority of models; an overly pristine pre-industrial may contribute to this (Menon et al., 2002; Carslaw et al., 2013). We

anticipate that the increase in PI sulfate following the expansion of DMS chemistry (**Figure 11a**) may dampen the IRF in our simulations. We evaluate IRF by calculating changes in the SW cloud radiative effects (ΔCRE) from PI to PD conditions, following previous studies (Gettelman et al., 2019; Ghan, 2013). Our estimates of IRF encapsulate not only the Twomey and the Albrecht effect (Twomey, 1977; Albrecht, 1989), but also cloud feedbacks in response to meteorological changes driven by different

sulfate aerosol loadings. For instance, our simulations do not fix air temperature and wind fields. Deeper mixing and stronger turbulence may affect cloud microphysics and limit/promote the precipitation efficiency of clouds, and hence alter cloud lifetime (Gettelman and Sherwood, 2016). These cloud





feedback mechanisms are still poorly constrained and contribute to the uncertainty in CRE estimations in both PI and PD eras.

600        The global-mean $\Delta$CRE estimated by differencing [STD_2000] and [STD_1850] is $-2.2$ W m$^{-2}$, comparable to a previous study using CAM6 (Gettelman et al., 2019) which reported SW $\Delta$CRE of $-2.1$ W m$^{-2}$ with CMIP6 emissions. They also showed that under CMIP5 emissions the magnitude of the SW $\Delta$CRE drops to $-1.4$ W m$^{-2}$, closer to the AR5 range. Our expanded DMS chemistry leads to a modest (~5%) strengthening in the simulated IRF ($-2.3$ W m$^{-2}$). However, due to the high variability of the cloud effects, these differences, both in the global values and the zonal means shown in **Figure 12c** are not statistically significant.

**a**

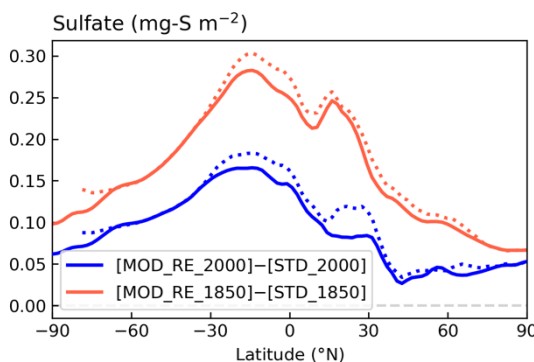



**b**                                                  **c**

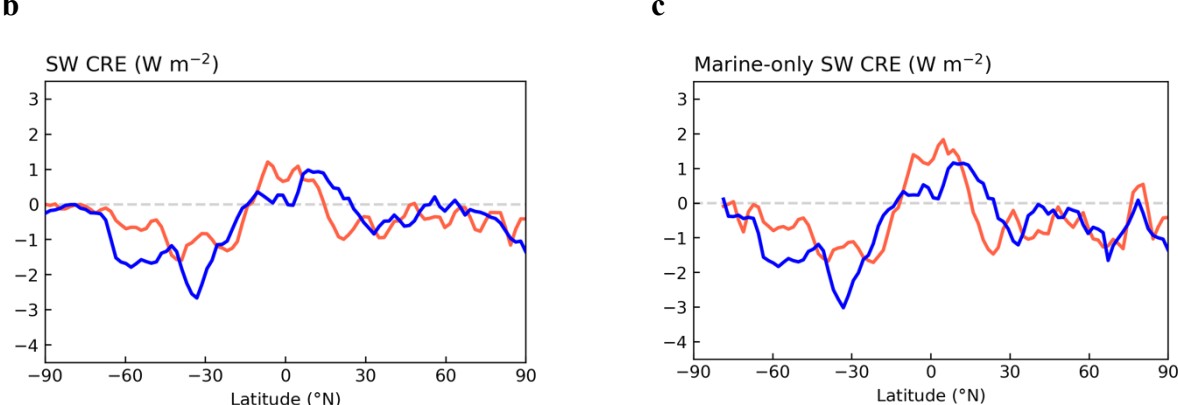

**Figure 12.** Contrasting the zonal-means of changes in **(a)** sulfate column concentration, **(b)** SW CRE, and **(c)** marine-only SW CRE modeled with [MOD_RE] and [STD] chemistry for PI and PD simulations. Dotted lines in **(a)** show the zonal mean marine grid cells only.

The strengthening of the IRF is opposite in sign to the expected response to the increase in PI
sulfate. Carslaw et al. (2013) describe how cloud albedo is much more strongly sensitive to CCN in the PI, suggesting that higher PI aerosol burden may decrease the cloud response to anthropogenic increases. **Figure 13** illustrates the change of SW CRE and sulfate burden from multiple simulations from PI to PD. Counter to expectations, we find that the SW CRE is more sensitive to each unit increment of sulfate burden (steeper slope) when the PI aerosol burden is higher (in [MOD_RE], shown in dark red)
compared to our baseline simulations ([STD], shown in blue). **Figure S10** shows that the of PI-to-PD change in CCN and cloud properties are also more sensitive to the change in sulfate burden in [MOD_RE] than [STD]. Hence, each unit PI-to-PD increase in sulfate burden in [MOD_RE] appears to induce more numerous and smaller cloud droplets and enrich cloud water content, enhancing cloud albedo. This could contribute to the enlarged change in SW CRE (or the IRF) in [MOD_RE] even
though its PI-to-PD sulfate burden increment is smaller than [STD].

This sensitivity may also be related to the spatial distribution in the change in DMS-derived sulfate burden. **Figure 12** shows that the increase in sulfate burden from PI to PD is stronger in the





marine atmosphere, as expected. In a sensitivity test, [GAS_RE], where only the gas-phase reactions of the new DMS oxidation scheme are enabled but not the aqueous-phase reactions (with the exception of

aqueous phase oxidation of $SO_2$), produces a less negative IRF of 1.7 W m$^{-2}$. **Figure S11** illustrates that this sensitivity simulation follows the anticipated response, with an increase in PI sulfate decreasing the change in liquid water path (LWP) from PI and PD (particularly in low clouds) and thus dampening the IRF, indicating the important climate implications of marine stratocumulus clouds (Wood, 2012). Contrasting [MOD_RE] and [GAS_RE] reveals that the introduction of the aqueous-phase pathway

contributes to large changes in the PD-PI sulfate co-located with high clouds over the tropics and low clouds over the Southern Ocean. Though the change in sulfate in both regions produce stronger regional cooling IRF, this appears to be the result of two different processes. In the tropics, decreases in sulfate in the presence of high clouds modify ice nucleation (Gettelman et al., 2010), leading to increased ice water path (IWP) and strengthened cloud cooling over the tropical oceans. Over the Southern Ocean,

decreases in sulfate attributable to the aqueous-phase chemistry are associated with even higher LWP in low clouds, further exaggerating the local cooling IRF compared to the gas-phase only simulation. Even though the amount of sulfate produced by the aqueous-phase pathway is relatively small (~8% of DMS-derived sulfate), it appears to have a disproportionate impact on clouds and the estimate of aerosol IRF, suggesting a strong sensitivity of cloud properties to the spatial distribution of natural marine

sulfate. More work is needed to better understand this response.

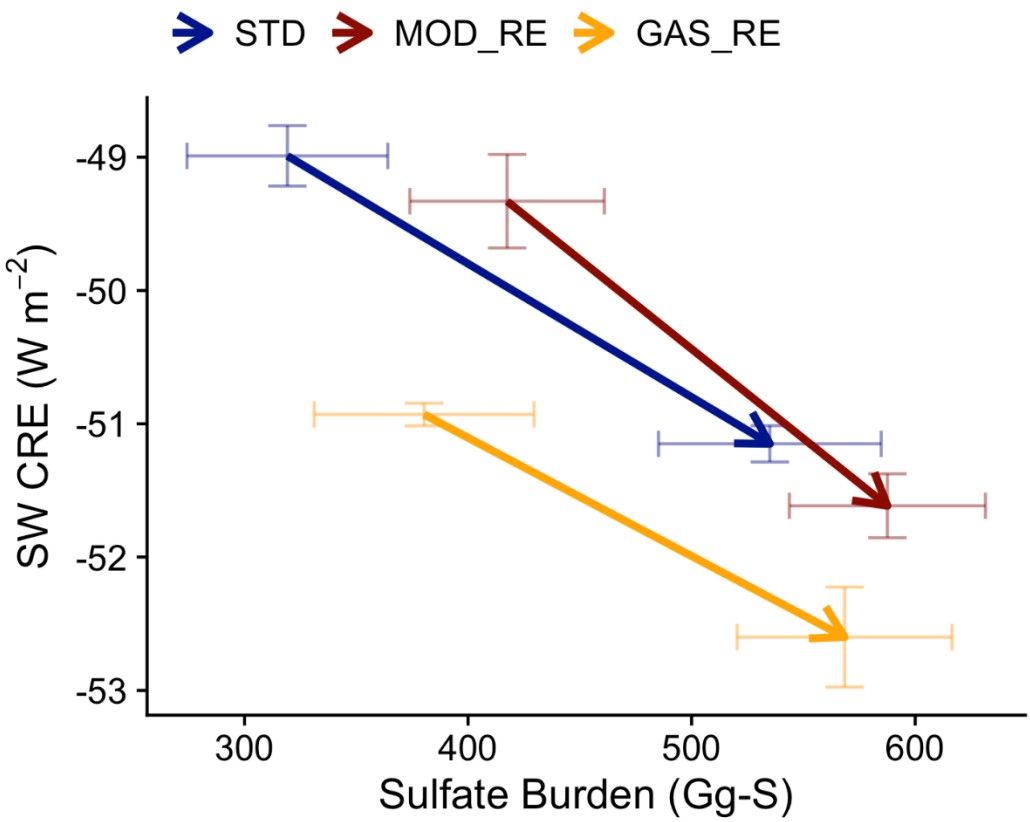

**Figure 13.** PI-to-PD changes in SW CRE and sulfate burden of simulations in this study. [STD] (blue) refers to the simulation with model default chemistry. [MOD_RE] (dark red) denote the cases with our expanded DMS chemistry implemented with all gas-phase, aerosol-phase, and in-cloud reactions. [GAS_RE] (yellow) which only includes the expanded gas-phase reactions is also shown. Arrows indicate changes from PI (tails) to PD (heads). Horizontal and vertical error bars span the 1-σ variabilities of burden and SW CRE in the simulations, respectively.



# 5. Conclusions

We expand the chemical mechanism in CAM6-chem to better describe DMS oxidation in the atmosphere, determine the formation of the intermediate sulfur products, and estimate the aerosol radiative implications under the PI and PD periods.

Uncertainty in our estimate of sulfate response to the new DMS chemistry is largely associated with estimated reaction rates. Some rate constants for the multiphase reactions are obtained from a limited set of box-model and laboratory studies which have not been validated with field measurements. For example, our rate constant for $MS^-_{(aq)}+OH_{(aq)}$ from Zhu et al. (2003) is 4.7 times smaller than another lab study (Milne et al., 1989), potentially leading to a higher global tropospheric MSA burden by ~30% (Chen et al., 2018). As discussed in **Sect. 3.2**, it is likely that our model overestimates the concentration of particulate MSA and underestimate gaseous MSA when compared with in situ measurements (e.g., ATom, and Yan et al. (2019) in the Southern Ocean. This could result in an overestimate of sulfate given that gas-phase MSA is expected to have a longer lifetime than particulate MSA and $H_2SO_4$ vapor (Berresheim, 2002).Our comparisons with observations also suggest that emissions of DMS, in particular a likely overestimate over the Southern Ocean, play an important role in dictating the regional loading of secondary oxidation products

This study included a relatively new chemical mechanism for the formation and loss of HPMTF. The rate of isomerization of MSP ($k_{iso}$) controls the production of HPMTF. The analyses reported here are based on an observationally constrained value of $k_{iso}$ (0.04 s$^{-1}$ at 293K) (Veres et al., 2020), which is slower than previously experiment- and model-based estimates of 0.23–2.1 s$^{-1}$ (Wu et al., 2015; Berndt et al., 2019). We find that a faster $k_{iso}$ (0.12 s$^{-1}$ at 293K based on Ye et al. (2021) has a negligible impact on the HPTMF burden (+4.1%) and resulting sulfate formation. However, the HPMTF burden is quite sensitive to the loss of HPMTF due to cloud uptake ($k_{HPMTF+cloud}$), which was recently suggested as a particularly important sink of HPMTF in the MBL (Veres et al., 2020; Vermeuel et al., 2020). These large changes suggest that further field measurements are needed to better understand the cloud uptake process of HPMTF and the resulting formation of in-cloud sulfur products.



In this study, we dramatically expand the DMS oxidation mechanism within an Earth System Model. Doing so increases the global sulfate burden by 8.8% in PD, and 29% in PI. While we anticipated that a larger PI burden of sulfate would dampen the aerosol IRF, our simulations instead suggested that the role of aqueous phase chemistry, though modest in terms of the sulfate burden,

confounds this effect. In a simulation with only updated gas phase chemistry, the increased PI burden decreased the IRF as anticipated ($-2.2$ W m$^{-2}$ in standard chemistry vs. $-1.7$ W m$^{-2}$ with updated gas-phase chemistry). However, high clouds in the tropics and low clouds in the Southern Ocean appear to be particularly sensitive to the sulfate produced via the aqueous phase pathway, counteracting the effect of the additional sulfate formed via the gas-phase pathways (net $-2.3$ W m$^{-2}$). These large differences

confirm the high sensitivity of aerosol indirect effects to the natural aerosol background, while revealing complex cloud responses to aerosol produced in different geographical regions via different pathways. More work is needed to understand these responses (e.g., better understanding of cloud responses to aerosols formed in the aqueous phase, observational constraints on the cloud uptake process of HPMTF via both laboratory and field measurements). While our new chemistry increases computational costs

by ~15% (in core-hours/simulation-year), this study suggests that a detailed description of the chemical oxidation of DMS and its products, and particularly the chemistry relevant to pristine conditions, is needed to accurately represent the abundance of natural sulfur species in the marine atmosphere and changes in natural aerosol burden over time.



**Code availability**

The modified codes of CESM2 developed in this study will be available upon request.

**Author Contribution**

KMF, CLH and JHK formulated the overarching research goals and aims. KMF and CLH designed the
methodology. KMF implemented the new codes into CAM6-chem based on the standard model
developed by SW, DJ, AG, ZL, XL, and RAZ. KMF validated the model results against observational
data provided by EA, DRB, JLJ, PC, PV, TSB, JES, and MZ. KMF analyzed the data and created
figures. KMF and CLH wrote the initial draft of this manuscript. All authors reviewed this manuscript.

**Competing Interests**

The authors declare that they have no conflict of interest.

**Acknowledgment**

This work was supported by U.S. Department of Energy (DOE) (Ref: DE-SC0018934) awarded to CLH
and JHK. RAZ acknowledges support by the U.S. DOE Office of Science, Office of Biological and
Environmental Research (BER), Earth and Environmental System Modeling (EESM) program as part
of its Earth System Model Development (ESMD) activity. JLJ and PC were supported by the funding
from NASA (Ref: 80NSSC18K0630 & 80NSSC19K0124). JES and MAZ were supported by ARM
and DOE's Atmospheric System Research, an Office of Science Biological and Environmental
Research program. We thank Rebecca Schwantes, Simone Tilmes, and Louisa Emmons for their





contribution to the modeling of this study. This material is based upon work supported by the National Center for Atmospheric Research, which is a major facility sponsored by the NSF under Cooperative Agreement No. 1852977. The high-performance computing was conducted on Cheyenne (doi:10.5065/D6RX99HX) provided by NCAR's Computational and Information Systems Laboratory,

sponsored by the National Science Foundation. We also thank Andy Neuman, Alan Bandy, Barry J. Huebert, and Stephen R. Springston for their contribution to the measurements used in this study. ACE-ENA data were obtained from the Atmospheric Radiation Measurement (ARM) User Facility, a U.S. DOE Office of Science user facility managed by the Biological and Environmental Research Program.



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
