# Peer review of "Exploring DMS oxidation and implications for global aerosol radiative forcing"

_Atmospheric Chemistry and Physics, 2021_

## Author Response (AR1)

**Responses to the Review 1 (RC1)**

Comments:

The authors report an expanded scheme of the widely used CAM-Chem model to reconsider atmospheric DMS chemistry in a more detailed way. Observations of the new and existing scheme are compared to observations. The implications for natural aerosols and resultant radiative implications are then considered. The conclusions for sulfur chemistry in the Southern hemisphere and the importance of considering loss routes via clouds is notable.

The paper is well written, well structured, and broadly covers some questions the community has had since recent lab and field papers on novel atmospheric sulfur chemistry (i.e. HPMTF). I would recommend publication after considering the minor points and suggestions below.

Reponses:

We thank the reviewer's positive and constructive feedback. We have addressed and revised the manuscript according. Our point-to-point responses are provided below.

Comments:

Line 38 - Please expand the acronym "VAMOS"

Reponses:

VAMOS here stands for "Variability of the American Monsoon Systems". The acronym is now explained in this updated sentence in the abstract:

"The expanded scheme improves the agreement between modeled and observed concentrations of DMS, MSA, HPMTF, and sulfate over most marine regions based on the NASA Atmospheric Tomography (ATom), the Aerosol and Cloud Experiments in the Eastern North Atlantic (ACE-ENA), and the **Variability of the American Monsoon Systems (VAMOS)** Ocean-Cloud-Atmosphere-Land Study Regional Experiment (VOCALS-REx) measurements."

The acronym is also detailed on lines 417-418 in the updated text.

Comments:

Table 2 - Is 2000 a typical year to fix volcano emissions? Or a low, medium or high emission year?

Responses:

For continuous outgassing volcanic emissions of $SO_2$ and primary sulfate, we follow the configuration described in Emmons et al. (2020), which assumed a constant rate in both PD and PI and is based on the GEIA inventory.

For eruptive volcanic $SO_2$ emissions, we are using the average emission rates between 1995–2005 to ensure an average impact of such emissions. The same emissions are used for both PD and PI periods, such that differences in emissions of $SO_2$ and primary sulfate between the PD and PI simulations are only attributable to anthropogenic sources.

We revised the last paragraph of Section 2.2 to describe our configurations with details:

"Volcanic emissions are fixed at the same level in both PI and PD simulations. **Emissions from continuously outgassing volcanos are constant (97.5% as SO2 and 2.5% emitted as primary sulfate aerosols) based on the GEIA inventory (Andres and Kasgnoc, 1998). We use time-averaged (1995–2005) eruptive volcanic emissions of SO2 to impose an average forcing from volcanic eruptions reaching the stratosphere, derived from the database of Volcanic Emissions for Earth System Models (VolcanEESM), version 3.10 (Neely and Schmidt, 2016).** $SO_2$ emissions from aircraft (up to ~15 km) and $SO_2$ & primary sulfate emissions from volcanos (up to ~30 km) are considered as elevated emissions while other sources of $SO_2$ emissions and oceanic DMS emissions are at the surface. A breakdown of SO2 emissions in this study is summarized in **Table 2**."

Ref.: Emmons, L. K., Schwantes, R. H.,Orlando, J. J., Tyndall, G., Kinnison, D., Lamarque, J. F., et al. (2020). The Chemistry Mechanism in the Community Earth System Model version 2 (CESM2). Journal of Advances in Modeling Earth Systems, 12, e2019MS001882. https://doi.org/10.1029/2019MS001882

Comments:

Figure 1 - please consider rotating the figure 90 degrees, so have water to one side. Having the "water" at the bottom of the figure may confuse readers into thinking that sea-surface reactions are being considered, rather than aqueous-phase reactions in aerosols and cloud droplets.

Reponses:

We appreciate the reviewer's suggestion and agree that the figure could be further improved. Instead of rotating it, we have made the following amendments on **Figure 1**:

1) We rephrased "air" and "water" in the figure to "gaseous" and "aqueous", respectively.
2) We replaced the shape-cornered rectangular shape with a round-cornered one to reduce the possible sense of "air-sea" contrast.
3) We emphasized in the caption of **Figure 1** that it's a schematic describing the atmospheric chemistry of DMS oxidation.

Please see the revised Figure 1 and the revised sentence in the caption as below:

[Figure]

The first sentence in the caption of **Figure 1** now reads:

"A schematic summary of our expanded **atmospheric** chemistry of DMS oxidation in CAM6-chem…"

Comments:

Lines 300-315 - Please consider adding a comment on the comparability of modelled oxidants with other models (e.g. global tropospheric average values for NO3, Cl, and BrO), or the certainty of modelled values used here and agreement with observations. How much could a difference in this model's predicted fields impact the notability of a specific route? For instance, Cl & BrO show large differences between studies even within the same models or even observational techniques as some of the authors of this paper have recently reported [Wang et al 2021].

Wang, Xuan, Daniel J. Jacob, William Downs, Shuting Zhai, Lei Zhu, Viral Shah, Christopher D. Holmes et al. "Global tropospheric halogen (Cl, Br, I) chemistry and its impact on oxidants." Atmospheric Chemistry and Physics Discussions (2021): 1-34.

Reponses:

We appreciate the suggestions. A detailed comparison is beyond the scope of this study. However, we have added text after the original Lines 300–315, to note that large discrepancies in modeled and observed Cl & BrO have been reported in recent study and warrant further investigation.

"**We note that recent studies (e.g., Wang et al., 2021b) have shown that large discrepancies in Cl and BrO are found within the same models and/or sets of measurements. Further investigation of how uncertainties in the representation of the halogen cycle feed back onto DMS chemistry is hence warranted.**"

Comments:

Line 416-418 - has the impact of meteorological variability been tested on the specific runs used here? Why not just use a single year (e.g. 2000) meteorology for all runs?

Reponses:

We used free-running dynamics instead of fixed-year meteorology to allow clouds to interact with the changing DMS-derived aerosols in the model so that we could capture the aerosol indirect effects.

We found that the impact of meteorological variability as measured by the interannual variability of global annual mean burden of the sulfur species are generally small, e.g., as mentioned in Lines 342–343 in the initial submission: "The PD global annual mean burden for sulfate aerosol is 582 Gg-S in [MOD_2000], with an interannual variability of 46 Gg-S (standard deviation of annual means)."

**Responses to the Review 2 (RC2)**

Comments:

Fung et al present a model study of the global DMS oxidation system, extending their chemistry scheme to take into account new insights around HPMTF. The revised chemistry scheme also includes MSA chemistry, knowledge of which is somewhat established but seldom included in models. A comparison of the revised model output to recent measurements is also presented.

The updated chemistry is used in calculating a revised aerosol indirect radiative forcing, arriving at a value of -2.3 Wm-2. This value is similar to previous estimates from this model, but the updated chemistry reveals significantly altered contributions from gas- and aqueous-phase oxidation pathways, and associated spatial differences.

The manuscript is well written, and considers a range of uncertainties in the reaction rates etc, including their potential impacts.

The supplemental material contains additional valuable figures and information. Inclusion of this information in the main manuscript could easily be justified (the supplementary material is extensively referred to throughout the manuscript), but would make the manuscript considerably longer and would likely distract from the main points of the study.

I have two major comments, and recommend publication when the points below are addressed.

Reponses:

We thank the reviewer's positive and constructive feedback. We have addressed and revised the manuscript according. Our point-to-point responses are provided below.

Comments:

**Major comments**
DMS emissions are suggested to be too high, leading to too high concentrations being simulated (when compared to observations). A reduced DMS flux simulation is provided in the supplementary material, leading to better agreement with observed DMS. What impact does the reduced DMS flux have on MSA, HPMTF, IRF etc? Presumably the burdens of these species are significantly altered. Interpretation of Figure S5 might also differ. All of this suggesting there would be consequences for aerosol IRF if [DMS]a was more accurately simulated.

Reponses:

We appreciate this question. We performed the (somewhat arbitrary) sensitivity experiment for emissions shown in the supplementary materials for one year only. To accurately capture the response, particularly on IRF, would require an additional 10 years of simulation, which we do not believe would be a good use of computational resources.

Based on the single year of simulation, we find the following: In the reduced DMS flux simulation, the PD global burden of HPMTF and MSA are lowered by 21% and 20% compared to [MOD_2000], respectively. The average surface concentration of HPMTF (shown in **Figure S5**) under the reduced DMS flux scenario could decrease by 55% in the Southern Ocean, compared to [MOD_2000]. With this decrease, a slower $k_{HPMTF+cloud}$ (< 5 x $10^{-5}$ $s^{-1}$) might be more appropriate to accurately capture the cloud uptake of HPMTF in this region. With the reduced DMS flux, the IRF estimated using the [MOD_RE] setting is negligibly different (~1.3% less negative) from the original [MOD_RE] simulation (–2.3 W $m^{-2}$).

Hence, our sensitivity simulation suggests that the uncertainty in DMS flux, could certainly alter sulfur burdens in the Southern Ocean region, but does not appear to impact the climate forcing estimates. Given that this response is not well characterized by this one-year simulation, we have added the following text to the conclusions in bold):

"Our comparisons with observations also suggest that emissions of DMS, in particular a likely overestimate over the Southern Ocean, play an important role in dictating the regional loading of secondary oxidation products**; the climate response to these changes should be further investigated.**"

We also expand the text on line 473-474 to read (new text in bold):
"Constraining DMS emissions is beyond the scope of this work, but is clearly a major source of uncertainty **that may impact the sulfur budget discussed in Sect. 3.1 and climate response discussed below.**"

Comments:

Has any evaluation been conducted with respect to aerosol number concentrations, e.g. CCN? How well are these quantities constrained? It is not currently possible to assess the ability of the model to reproduce fundamental aerosol parameters, which limits confidence in the conclusions. Please provide, at least, surface maps of CCN so that the reader can determine whether the model has an ability to simulate CCN effectively (and therefore CRE).

Reponses:

We welcome the reviewer's suggestion and have added a new **Figure S1 (i)** in the **Supplementary Information** to show the annual-mean surface concentration of CCN for our readers' reference:

[Figure]

**Minor comments**
There are several references to IPCC AR5, which itself refers to quite old literature. It would be better to refer to AR6. And better still, to the literature referred to therein.

Reponses:

We thank the reviewer for pointing out that the preliminary version of AR6 is out. Since the final edits of AR6 is not available at this moment, we prefer to keep the citation as is. However, we revised the beginning of the **Introduction** to note the preliminary release of AR6.

The revised sentence now reads:

"The IPCC AR5 (Myhre et al., 2013) **and the recent preliminary release of AR6 (https://www.ipcc.ch/report/sixth-assessment-report-cycle/)** indicate that atmospheric aerosol particles are a dominant source of uncertainty in global climate forcing."

Comments:

Is there any vertical distribution of the sulfur (or other) emissions? Or are all emissions injected at the surface?

Reponses:

There are both elevated and surface emissions. $SO_2$ emissions from aircraft (up to ~15 km) and $SO_2$ & primary sulfate emissions from volcanos (up to ~30 km) are considered as elevated emissions while other sources of $SO_2$ emissions and oceanic DMS emissions are at the surface. This is now detailed in the last paragraph in **Section 2.2**:

"**$SO_2$ emissions from aircraft (up to ~15 km) and $SO_2$ & primary sulfate emissions from volcanos (up to ~30 km) are considered as elevated emissions while other sources of $SO_2$ emissions and oceanic DMS emissions are at the surface.** A breakdown of $SO_2$ emissions in this study is summarized in Table 2. We use the same emissions for other species as the standard CMIP6 simulations (Emmons et al., 2020)."

Comments:

Fig.1 caption: it would be useful to have an indication of what is considered long-lived

Reponses:

We thank the reviewer for this suggestion and have revised the caption. The revised portion of the caption now reads:

"Key relatively long-lived species (DMS, MSA, HPMTF, $SO_2$, and sulfate) **with lifetimes of >0.5 days** are highlighted in bold."